# Seasonal Variation of Biogenic and Anthropogenic VOCs in a Semi-Urban Area Near Sydney, Australia

Jhonathan Ramirez-Gamboa [1,2,*], Clare Paton-Walsh [1,2], Ian Galbally [1,2,3], Jack Simmons [1,2], Elise-Andree Guerette [1,2], Alan D. Griffith [1,2,4], Scott D. Chambers [1,2,4] and Alastair G. Williams [1,2,4]

1   Centre for Atmospheric Chemistry, University of Wollongong, Wollongong, NSW 2522, Australia; clarem@uow.edu.au (C.P.-W.); Ian.Galbally@csiro.au (I.G.); js828@uowmail.edu.au (J.S.); elise-andree.guerette@csiro.au (E.-A.G.); alan.griffiths@ansto.gov.au (A.D.G.); szc@ansto.gov.au (S.D.C.); agw@ansto.gov.au (A.G.W.)
2   School of Earth, Atmospheric and Life Sciences (SEALS), University of Wollongong, Wollongong, NSW 2522, Australia
3   The Commonwealth Scientific and Industrial Research Organisation, CSIRO, Canberra, VIC 3169, Australia
4   Australian Nuclear Science and Technology Organisation (ANSTO), Lucas Heights, NSW 2522, Australia
*   Correspondence: jrg604@uowmail.edu.au

**Abstract:** Volatile organic compounds (VOCs) play a key role in the formation of ozone and secondary organic aerosol, the two most important air pollutants in Sydney, Australia. Despite their importance, there are few available VOC measurements in the area. In this paper, we discuss continuous GC-MS measurements of 10 selected VOCs between February (summer in the southern hemisphere) and June (winter in the southern hemisphere) of 2019 in a semi-urban area between natural eucalypt forest and the Sydney metropolitan fringe. Combined, isoprene, methacrolein, methyl-vinyl-ketone, α-pinene, p-cymene, eucalyptol, benzene, toluene xylene and tri-methylbenzene provide a reasonable representation of variability in the total biogenic VOC (BVOC) and anthropogenic VOC (AVOC) loading in the area. Seasonal changes in environmental conditions were reflected in observed BVOC concentrations, with a summer peak of 8 ppb, dropping to approximately 0.1 ppb in winter. Isoprene, and its immediate oxidation products methacrolein (MACR) and methyl-vinyl-ketone (MVK), dominated BVOC concentrations during summer and early autumn, while monoterpenes comprised the larger fraction during winter. Temperature and solar radiation drive most of the seasonal variation observed in BVOCs. Observed levels of isoprene, MACR and MVK in the atmosphere are closely related with variations in temperature and photosynthetically active radiation (PAR), but chemistry and meteorology may play a more important role for the monoterpenes. Using a nonlinear model, temperature explains 51% and PAR 38% of the isoprene, MACR and MVK variation. Eucalyptol dominated the observed monoterpene fraction (contributing ~75%), with p-cymene (20%) and α-pinene (5%) also present. AVOCs maintain an average concentration of ~0.4 ppb, with a slight decrease during autumn–winter. The low AVOC concentrations observed indicate a relatively small anthropogenic influence, generally occurring when (rare) northerly winds transport Sydney emissions to the measurement site. The site is influenced by domestic, commercial and vehicle AVOC emissions. Our observed AVOC concentrations can be explained by the seasonal changes in meteorology and the emissions in the area as listed in the NSW emissions inventory and thereby act as an independent validation of this inventory. We conclude that the variations in atmospheric composition observed during the seasons are an important variable to consider when formulating air pollution control policies over Sydney given the influence of biogenic sources during summer, autumn and winter.

**Keywords:** volatile organic compounds; biogenic; anthropogenic; GC-MS; seasonality

## 1. Introduction

Volatile organic compounds (VOCs) are a group of carbon-based gases emitted by biological and anthropogenic sources that are characterised by their high vapour pres-

sure at ambient temperatures [1–3]. Biogenic VOCs (BVOCs) are involved in biological signalling [4] and are also associated with changes to regional/global climate [5,6]. Anthropogenic VOCs (AVOCs) are important in urban environments, and in fire-prone regions of the world (such as Australia), vegetation fires can also be a large, although irregular, source of VOCs in the atmosphere [7–9]. Both BVOCs and AVOCs are important drivers of air quality since they are precursors in the formation of ozone [10,11] and fine particulate matter [12]. VOCs are oxidised, producing $RO_2$ radicals. These react with $NO_2$, generating ozone and recycling the NO into $NO_2$. At the same time, these oxidised products continue the reaction chain, producing secondary organic aerosols [13–15].

Although there are different sources of VOCs on the planet, most of the emissions are related to plants [4]. Plants emit hundreds of different BVOCs to the atmosphere, but isoprene is the most common [16]. Isoprene has been related to various essential processes in plant metabolism, including protection from parasites and herbivores, attracting pollinisers and other beneficial insects and preserving the organic structures in the leaves during stressful situations such as droughts, heatwaves or floods [17,18]. In atmospheric chemistry, isoprene is one of the precursors for ozone and aerosol formation in the atmosphere. Monoterpenes have been identified alongside isoprene as a group of BVOC species that have a significant role in the chemistry of the atmosphere [19]. There are many uncertainties associated with the flux estimation of these compounds, particularly because of the variety and different reactivities of the range of monoterpenes. Models and measurement campaigns have been used to estimate that, globally, the total flux of BVOCs to the atmosphere is between 500 and 750 Tg per year [20], with vegetation emitting around 90% of the non-methane VOCs in the atmosphere [4]. Isoprene and monoterpenes together comprise most of the global BVOC emissions and are some of the most important chemical species related to secondary organic aerosol and tropospheric ozone formation [21,22].

Despite only accounting for approximately 10% of global atmospheric VOC emissions [12], within urban environments, AVOCs are important for atmospheric chemistry, being precursors of both aerosol and ozone formation. One of the most studied groups of AVOCs is BTEX (comprising benzene, toluene, ethylbenzene and xylene) due to both its relatively large atmospheric abundance and the health risk potential of these chemicals [23]. BTEX is associated with emissions from combustion and evaporation in industries and vehicles [24]. BTEX has been shown to account for approximately 22% of total VOCs in the urban environment in Melbourne, with approximate fractional contributions of each component of BTEX being ~10% benzene, ~49% toluene, ~6% ethylbenzene and ~34% o, m and p-xylene [25].

Biogenic emissions of VOCs are most commonly estimated using models such as the Model of Emissions of Gases and Aerosols from Nature (MEGAN) [16], whilst AVOC emissions are most commonly estimated using regional or global emissions inventories such as the Emission Database for Global Atmospheric Research (EDGAR) [26]. Despite the refinement of these techniques over recent years, large uncertainties remain in estimated emissions of BVOCs and AVOCs in many parts of the world [27–32], with Australia a particularly poorly characterised region [33,34].

Southeastern Australian forest areas are characterised by the presence of multiple eucalyptus species. These species are counted as the most prominent in Australian forests, especially in the southeastern part of the country. Some eucalyptus species are among the highest emitters of VOCs globally [35,36], and for this reason, BVOC emissions from southeastern Australia (and particularly those from the forests surrounding the Sydney basin) are modelled to be amongst the highest in the world [20]. The few previous studies undertaken in this region have provided some important findings. Maleknia et al. 2009 [37] showed that some local eucalyptus species increase their VOC emissions when under stress. Wounding the branches or leaves of *Eucalyptus sideroxylon* leads to the release of stored oils or defence BVOCs depending on the plant characteristics [38]. A similar study using *Grevillea robusta* leaf mulch and wood chips found high emission of oxygenated VOCs from the material up to 30 h after the wounding [39]. The Sydney Particle Study

showed that isoprene and monoterpenes were highly correlated with the formation of organic aerosols in urban areas during summer [40]. Guerette et al. 2019 [41] showed that isoprene levels were highly elevated during extreme heat events and that the ratio between isoprene and its oxidation products could indicate the influence of different sources during the "MUMBA" campaign [10,42].

Emmerson et al. 2016 [33] compared the observations of BVOCs made during the Sydney Particle Study and the MUMBA campaign to a chemical transport model that incorporated the MEGAN emissions inventory [16,20]. The observed concentrations of isoprene and monoterpenes during those campaigns [42,43] were within a factor of two of each other, suggesting unusually high monoterpene amounts in the region. The modelled isoprene concentrations were up to a factor of six too high compared to the observations, and monoterpene concentrations were underestimated by a factor of up to four times [33]. This discrepancy highlights the large uncertainty in BVOC emissions from southeastern Australian forests and the need for further measurement campaigns to better characterise BVOC emissions in the region.

Emmerson et al. (2018) [44] compared MEGAN to the Australian Biogenic Canopy and Grass Emissions Model (ABCGEM) from CSIRO using in situ measurements of monoterpenes in rural and urban areas in Sydney. They found that the ABCGEM model has a better representation of local vegetation monoterpene emissions during high-temperature periods. Different factors, such as the emission factors or the leaf area index, used in the models affect the modelling results, but the emission dependence on light and temperature is key to predicting diurnal cycles. In MEGAN, monoterpene emissions depend both on light and temperature, whereas the ABCGEM model only includes a temperature dependence. The ABCGEM gives a better representation of local overnight monoterpene concentrations, implying that monoterpene emissions "from Australian vegetation may not be as light dependent as vegetation globally" [44]. Nevertheless, these conclusions are based on a very limited set of BVOC measurements in the region and further observations are needed to build confidence in our understanding of BVOC emissions in the Sydney basin.

The Department of Primary Industry and the Environment (DPIE) in the Australian state of New South Wales (NSW) provides an emissions inventory for NSW which is updated every 5 years. The NSW inventory is focused on anthropogenic emissions and uses AVOC emission factors from different international sources including the US EPA and the European model Computer Programme to calculate Emissions from Road Transport (COPERT, Queensland, Australia) [45]. Few measurement datasets of ambient atmospheric concentrations of AVOC exist in Australia that can be used to validate the NSW emissions inventory and so the accuracy of the emissions inventory remains uncertain. Recently, Smit et al. 2019 [46] and Smit et al. 2017 [47] measured AVOCs on-road and in a road tunnel. They found differences between the modelled AVOC chemical speciation profile in COPERT and the observations in both studies, showing an overestimation of alkanes and alkenes while underestimating alcohols. Further studies are needed to confirm these results and reduce the uncertainty in emissions estimates from Australian anthropogenic sources.

A project was designed to address a number of the observational gaps outlined above and was named COALA (Characterizing Organics and Aerosol Loading over Australia). COALA included a major international campaign over the Austral summer of 2020 at a site within a eucalypt forest near Cataract Dam, NSW, 34°14′ 41.0″ S,150° 49′ 24.1″ E in January–March 2020 focused on biogenic emissions. This was preceded by the study reported in this paper, known as the Joint Organic Emissions Year-round Study (or COALA-JOEYS), which aimed to characterise ambient concentrations of key VOCs and their seasonal variability in the region, so that the main campaign could be assessed in the context of the annual cycles. A semi-urban site was identified approximately 31 km southwest of Sydney, at Lucas Heights, NSW, situated between natural eucalypt forest and the Sydney metropolitan fringe. COALA-JOEYS was originally planned to cover a full annual cycle but was curtailed due to resourcing and logistical reasons. Nevertheless, the study provides a valuable dataset

of hourly observations of ambient atmospheric concentrations of key BVOCs and AVOCs during three seasons in 2019.

COALA-JOEYS provides evidence of the changing composition of BVOCs and AVOCs in the atmosphere from February (summer in the southern hemisphere) to June (winter in the southern hemisphere) during 2019 in this under-sampled region of southeastern Australia and can therefore help us to understand how these VOCs change with the seasons.

## 2. Experiment

### 2.1. Sampling Site

The sampling site for COALA-JOEYS was located at the Australian Nuclear Science and Technology Organisation (ANSTO), at Lucas Heights, New South Wales (34°03′ 09.4″ S 150°59′ 08.2″ E). The site was selected for its proximity to large areas of native forest, particularly to the north-northwest and southwest (Figure 1). Lucas Heights is located on the edge of the Sydney Basin, 31 km from the centre of Sydney. The site has an annual mean rainfall of 1142 mm, average summer maximum temperature of 26 °C and average winter maximum temperature of 15 °C. A municipal waste management facility is located 2 km to the northwest, the Engadine residential area is located to the east and a road carrying medium traffic density (the A6) surrounds the sampling area. Meteorological parameters were measured in a 50 m height tower with sensors at 10 and 50 m approx. ~500 m from the sampling site.

### 2.2. Sampling and Analysis System

The sampling inlet was installed on a mast on the second-floor balcony of a three-storey building, 7.5 m above ground level and 2 m above the balcony. The inlet consisted of polytetrafluoroethylene (PTFE) tubing of 3/8′ outside diameter, fitted with an upside-down PTFE beaker as a rain cap (Supplementary Figure S1A). The inlet line was 25 m long. Line exposed to direct sunlight was covered in garden hose in order to prevent photochemistry inline. Ambient air samples were drawn through the line at 12 L/min by a PTFE-coated diaphragm pump (N810 LABOPORT, KNF Neuberger, Freiburg, Germany). The inlet line flow was greatly in excess of the flow rate to the instrument, which transferred sample via a short (~1 m) length of 1/8″ PTFE tubing to the instrument at a flow rate of 100 mL/min. A check valve was installed downstream of the pump to prevent backflow. A 5-μm paper filter in a PTFE cartridge was installed inline to protect the pump from atmospheric particulates. The filter was changed on a quasi-monthly basis.

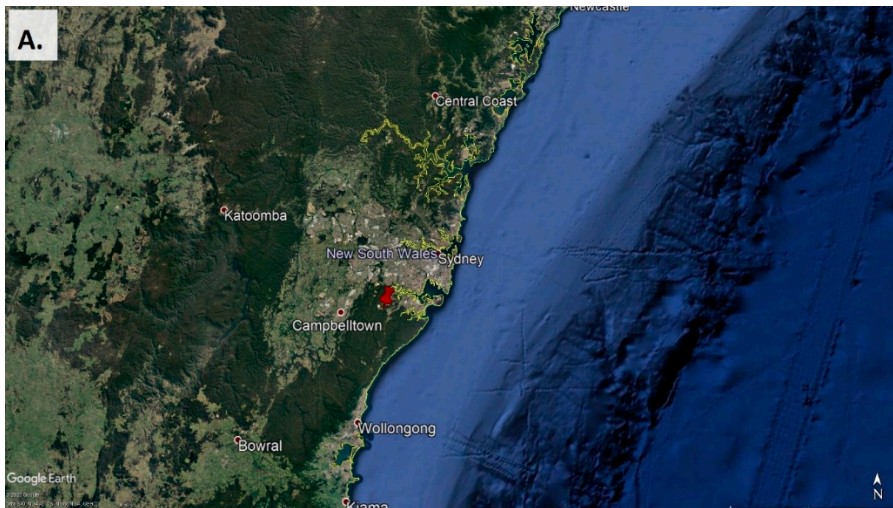

**Figure 1.** *Cont.*

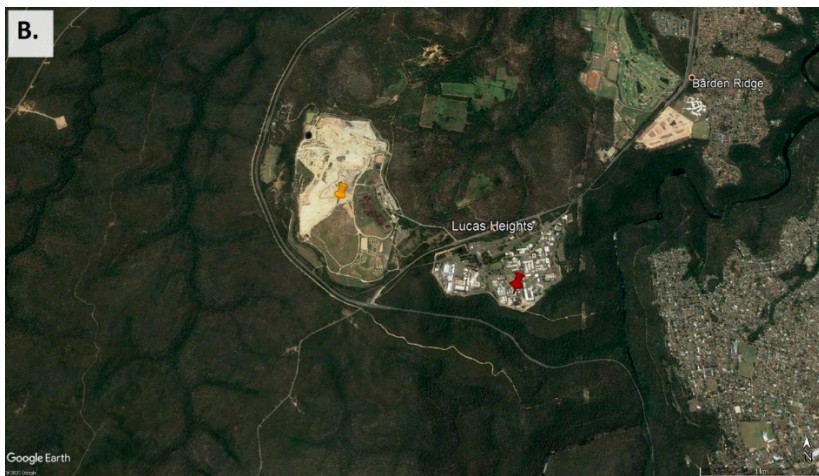

**Figure 1.** (**A**) Map of the sampling site location within its regional context in the southeast of Australia, showing ~100 km around the sampling point [48]. (**B**) Close-up view of the sampling site (red indicator) and its surroundings [49]. The yellow indicator shows the location of the Lucas Heights waste management facilities. Forest surrounds most of the area. There are residential areas close by to the east and further away to the west.

The integrated system including inlet, analytical equipment and auxiliary items is called the Biogenic Ambient Air Sampling System (BAASS) and is capable of continuous measurement of VOC concentration in ambient air with a one-hour time resolution using gas chromatography coupled to mass spectrometry or flame ionisation detection (GC/MS-FID). There were no point sources of VOCs in the immediate surroundings of the sampling site, but there was a nearby parking area. Quality control during the data processing identified two data points in the hourly measurements as AVOC outliers and these data were excluded from the analysis reported in this paper.

### 2.2.1. Water Trap and Thermal Desorption

Ambient air samples were analysed for key VOCs (see Table 1) using a thermal desorption–gas chromatography–mass spectrometry system (TD-GC-MS), pictured in Supplementary Figure S1B. The system consisted of three parts. An inlet manifold with eight inlet ports (Air Server, Markes International Ltd., Llantrisant, UK) allowed the system to switch from sampling from the inlet to sampling calibration gases or laboratory air. From the inlet manifold, samples passed through a Peltier-cooled glass water trap (KORI-xr, Markes International Ltd., Llantrisant, UK). This water trap, held at −20 °C during sampling, removed water from humid ambient samples by freezing. The water trap was heated to 300 °C and flushed with 50 mL/min ultra-high purity nitrogen between samples to vent accumulated water. Finally, the dried ambient air sample passed to a VOC trap (type U-T11GPC-2S, Markes International Ltd., Llantrisant, UK) held at −30 °C during sampling. After the trapping phase was complete, the VOC trap was heated at 40 °C/s and purged the sample to the GC for 5 min. Following sample purging, the VOC trap was further purged with 50 L/min dry carrier gas for 2 min to prevent contamination in the following samples. Supplementary Figure S1C shows a general diagram of the system.

**Table 1.** Selected ions used to detect targeted compounds in SIM mode during the sampling period. The target ion was used to estimate concentrations and uncertainties. The qualifier ions (1 and 2) were used to confirm the identity of the compound in the chromatogram. Retention times (in seconds) are for an initial oven temperature of 35 °C. Ions were selected using the NIST library [50].

| Compound | Boiling Point (°C) | Retention Time (s) | Target Ion | Qualifier 1 | Qualifier 2 |
|---|---|---|---|---|---|
| Isoprene | 34.07 | 198 ± 20 | 67 | 68 | - |
| Methacrolein | 68.4 | 224 ± 15 | 70 | 39 | 41 |
| MVK | 81.4 | 240 ± 15 | 55 | 70 | - |
| Benzene | 80.1 | 325 ± 30 | 78 | 77 | - |
| Toluene | 110.6 | 500 ± 25 | 92 | 91 | - |
| p-xylene | 138.4 | 687 ± 15 | 106 | 91 | - |
| α-pinene | 156.1 | 798 ± 10 | 93 | 136 | 121 |
| TMB-1,2,4 | 169.3 | 895 ± 10 | 105 | 120 | 119 |
| p-cymene | 176.8 | 951 ± 10 | 119 | 134 | - |
| Eucalyptol | 176.4 | 965 ± 10 | 81 | 154 | 139 |

### 2.2.2. Gas Chromatography–Mass Spectrometry

Analytes were separated using a 7890B Agilent Technologies Gas Chromatograph equipped with an HP-5 MS column (30 m × 0.25 μm × 0.25 mm ID; Agilent Technologies, Santa Clara, CA, USA). The column temperature was held at 35 °C for 5 min before ramping at 4 °C/min to 160 °C. The ramp was then increased to 30°/min to 280 °C with a final hold time of 1 min. Helium was used as a carrier gas to the GC. A split of 20 mL/min was used with a column flow of 1 mL/min, giving a split ratio of 20:1. The eluent was detected using a single quadrupole electron ionisation mass spectrometer (Agilent 5977B with Xtr EI source, Agilent Technologies, Santa Clara, CA, USA). The mass spectrum detector was operated in selected ion mode (SIM). Details of SIM windows are presented in Table 1. The starting column temperature was changed from 35 to 30 °C in mid-April. Before this, the oven was taking too long to reach 30 °C due to periodic failures in the air-conditioning unit in the laboratory, losing several sampling hours per day. The change in temperature did not affect the operation of the instrument except to change the elution time of the compounds by ~40 s. Details of the inlet losses, humidity effect, linearity and contamination can be found in the Supplementary Materials.

### 2.3. Sampling Regime

Ambient samples were collected at 100 mL/min for 30 min, giving a total sampled volume of 3000 mL. Calibration samples consisting of a mixture of ten targeted species from a calibration cylinder (Air Liquide, Houston, TX, USA; see Supplementary Table S1) were collected at 3 mL/min to a total sampled volume of 5 mL. Blank measurements were defined as measurements of ultra-high-pure (UHP) nitrogen collected at 100 mL/min for 30 min, giving a total sampled volume of 3000 mL, identical to the sample. Measurements were taken on a quasi-hourly basis. Sampling was near-continuous for the duration of the experiment, though large gaps exist in the measurement timeseries. These gaps are associated with system maintenance and characterisation, or exhaustion of supply gases. The sampling schedule varied throughout the campaign. A calibration measurement was taken every six to twelve hours, with blank measurements occurring before calibrations and a "flush" sample after calibrations to remove the carryover effect (see Supplementary Materials).

### 2.4. Blank Selection

UHP nitrogen was used as a blank for the system. Isoprene, methyl-vinyl-ketone, methacrolein, eucalyptol, alpha-pinene and tri-methyl-benzene did not show any response in the blank samples. The blank concentrations obtained for the other species are presented in Table 2. To confirm that UHP nitrogen blanks were appropriate, two alternative blanks were tested. These were (a) a zero air blank and (b) a blank where ambient air passed through a platinum-coated glass wool catalyst [51] that destroyed VOCs. Both types were collected at 100 mL/min for 30 min. The results were compared to the nitrogen blank samples. No significant differences were found between the three blank methods. UHP nitrogen was therefore used for most of the sampling period.

**Table 2.** Blank concentrations estimated from all UHP nitrogen samples during COALA-JOEYS.

| Compound | Mean Concentration (ppb) | Standard Deviation (ppb) |
|----------|--------------------------|--------------------------|
| Benzene | 0.01 | 0.003 |
| p-cymene | 0.05 | 0.018 |
| Toluene | 0.02 | 0.006 |
| p-xylene | 0.04 | 0.016 |

### 2.5. Concentration Estimation

Chromatograms were integrated to obtain the peak area of each targeted VOC using the GCWerks software [52]. The sensitivity of the instrument to each targeted compound was determined by weighting the nearest calibration point either side of the ambient measurement by the relative length of time between each calibration and the ambient sample. The results were exported, and the ambient concentrations for each VOC in turn were estimated using

$$C_{si} = \frac{V_{std}}{X_{i-j}A_{std\,j} + X_{i-k}A_{std\,k}} * \left( \frac{A_{si}}{V_{s\,i}} - \frac{A_b}{V_b} \right) * C_{std} \qquad (1)$$

where $C_{si}$ is the concentration of the sample $i$, $A_{si}$ is the peak area of the sample $i$, $X_{i-j}$ is the relative factor of time since the calibration $j$ was analysed with respect to sample $i$, $A_{std\,j}$ is the peak area of calibration $j$, $X_{i-k}$ is the relative factor of time since the sample was analysed with respect to calibration $k$, $A_{std\,k}$ is the peak area of calibration $k$, $V_{std}$ is the calibration volume sampled, $V_{si}$ is the ambient air volume sampled, $V_b$ is the blank air volume sampled, $A_b$ is the peak area of the blank, $C_{std}$ is the standard reported concentration.

### 2.6. Uncertainty and Limit of Detection

The uncertainty of the system was calculated using Equation (2), where $\Delta U^2$ is the total uncertainty, $\Delta \chi^2 Cal$ is the uncertainty of the calibration cylinders reported by the manufacturer, $\Delta \chi^2 analysis$ is the root mean square error of the running mean of the calibrations compared to the single calibrations. The scaling factor 4 represents the uncertainty of ambient measurements relative to the calibration uncertainty. This equation is used because there are no data on the repeatability of ambient measurements, so the repeatability of calibrations is used instead. The scaling factor is included to allow for extra uncertainty in ambient measurements. A detailed explanation can be found in the Supplementary Materials.

$$\Delta U^2 = 4 * \Delta \chi^2 analysis + \Delta \chi^2 Calibration\ std \qquad (2)$$

The limit of detection (LOD) of the instrument was estimated using Equation (3), where the signal to noise ratio (SNR) was calculated using the noise signal at the base of the chromatogram. The chromatogram section used for the LOD estimation had the highest

SNR ratio (higher noise) for each peak. The total uncertainty and LOD values are reported in Table 3.

$$\Delta U^2 = 4 * \Delta\chi^2 analysis + \Delta\chi^2 Calibration\ std \tag{3}$$

**Table 3.** Estimated LOD and relative uncertainty for hourly measurements of each VOC using the BAASS system during the sampling period.

| Compound | ΔX Cal. Standard | ΔX Analysis | Δ Total Uncertainty | LOD (ppt) |
|---|---|---|---|---|
| Isoprene | 0.05 | 0.11 | 0.23 | 9.6 |
| Methacrolein | 0.05 | 0.10 | 0.21 | 14.5 |
| Methyl-vinyl-ketone | 0.05 | 0.21 | 0.42 | 29.2 |
| Benzene | 0.05 | 0.09 | 0.19 | 12 |
| Toluene | 0.05 | 0.11 | 0.23 | 21.9 |
| p-xylene | 0.05 | 0.12 | 0.25 | 5.5 |
| α-pinene | 0.05 | 0.18 | 0.35 | 4.6 |
| TMB-1,2,4 | 0.05 | 0.11 | 0.22 | 12.9 |
| p-cymene | 0.05 | 0.12 | 0.24 | 4.7 |
| Eucalyptol | 0.05 | 0.35 | 0.70 | 29.1 |

### 2.7. Radon Observations and Atmospheric Mixing State Classification

Continuous hourly observations of atmospheric radon ($^{222}$Rn) concentration were made at 2 m above ground level using a 1500 L two-filter dual-flow-loop radon detector designed and built at ANSTO [53,54]. The operational characteristics, calibration and instrumental background correction methods for this detector have already been described by Chambers et al. (2011) [55]. Radon sampling was conducted approximately 300 m NW of the balcony from which the BAASS VOC observations were made.

Diurnal cycles of radon (see Figure 2) were typically characterised by a morning maximum (when atmospheric mixing was weakest) and a late afternoon minimum, when the atmospheric boundary layer (ABL) was deepest and most well-mixed. Peak nocturnal radon concentrations were a factor of 2–3 lower than reported elsewhere in the Sydney basin [56], primarily attributable to this site's location at the top of a ridge in complex topography (changes in elevation of ~180 m within a 1 km radius of the site). Under thermally stable nocturnal conditions, more common in autumn and winter, cold near-surface air can drain away from the ridge tops down into the surrounding valleys before local emissions have an opportunity to accumulate. This is evidenced by the approximately flat diurnal radon signals between 2 am and sunrise in April through June, rather than continued accumulation until sunrise (see Figure 2c). Since the source of BVOCs is the forest canopy, rather than the ground surface, this effect may be less pronounced for the BVOC observations, but likely still significant.

In this study, the radon observations have been primarily used to characterise the atmospheric mixing state. Since $^{222}$Rn is a naturally occurring, noble, radioactive gas with a horizontally distributed surface-only source, changes in its concentration on short (≤diurnal) timescales are a useful proxy for the behaviour of other atmospheric constituents with horizontally distributed near-surface sources. Radon's 3.8-day half-life ensures that it is a conservative tracer for typical mixing timescales in the ABL; however, this half-life is long enough for an air mass to retain a "memory" of influences of radon emission from nearby upwind source regions, referred to here as a "fetch effect". If the fetch contribution of an observed radon timeseries (red line in Figure 2a) can be approximated and subtracted from the observed concentrations, the resulting timeseries represents the atmospheric mixing-related influence on near-surface radon concentrations (see Figure 2b).

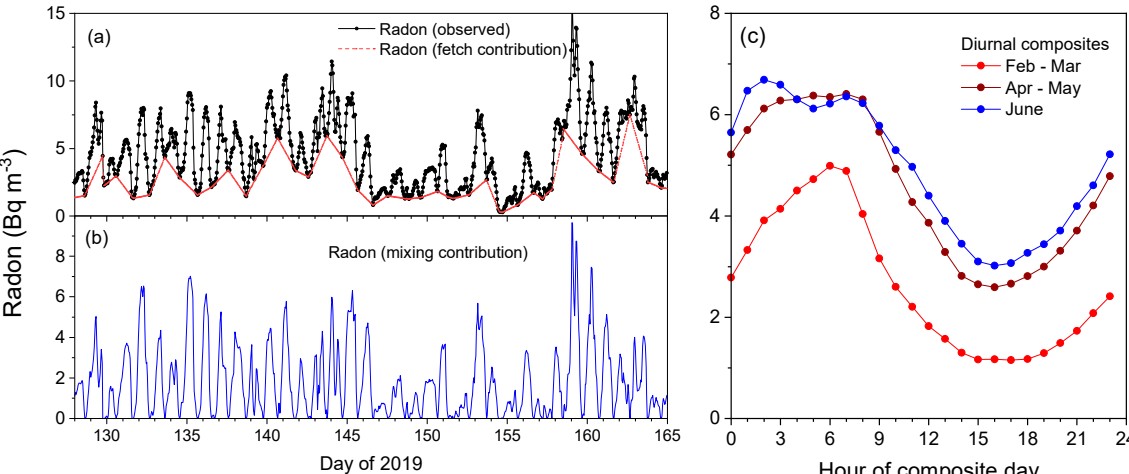

**Figure 2.** (**a**) A 5-week example of observed radon concentrations and the corresponding fetch-related contribution. (**b**) Mixing-related contribution to observed radon (observed—fetch), the *x* axis shows decimal days of 2019. (**c**) Diurnal composite radon concentrations for the hot (February–March), warm (April–May) and cold (June) months of the study.

As described in detail by Chambers et al. (2015, 2019) [56,57], the average nocturnal atmospheric mixing state can be approximated by (i) calculating the mean radon mixing component each night over a 10-h nocturnal window (19:00 h–05:00 h) referenced to the 19:00 h value, and (ii) grouping these nightly values each season into atmospheric mixing categories. At one extreme, nights on which there is only a small nocturnal mean radon mixing concentration are typically windy and overcast (near-neutral atmospheric mixing conditions), whereas nights with a high radon accumulation are typically calm with clear skies (most stable atmospheric mixing conditions). Occasions on which the nocturnal mean radon mixing concentration (referenced to the 1900 h value) reaches negative values are usually associated with a rapid air mass fetch change (synoptically non-stationary), as might be expected with the passing of a frontal system.

The nights were grouped into 4 broad nocturnal mixing categories, based on quartile ranges of the mean nocturnal radon mixing component: 1—near-neutral; 2—weakly stable; 3—moderately stable; 4—most stable and a fifth category for periods of nocturnal synoptic non-stationarity. Category 1 conditions were associated with the highest wind speeds, which were usually associated with advection of AVOC and BVOC contributions from regions quite remote from the sampling site.

### 2.8. Isoprene, Monoterpenes and Atmospheric Oxidation

A key issue in interpreting reactive VOCs in the atmosphere is their lifetime due to atmospheric oxidation. Estimates of the chemical lifetimes of the VOCs, $\tau_{chem}$, studied here drawn from the literature for standard conditions (1 atm and 25 °C) are presented in Table 4. Given the observed wind speeds, u, with a mean of 2.2 m/s and a typical extreme of 5.8 m/s, the distances that would be travelled in a straight line in one hour are approximately 7 and 22 km, respectively. Thus, for a VOC with a lifetime of $\tau_{chem}$, the observed concentrations would mainly be those emitted within the distance D given by

$$D = u * \tau_{chem} \tag{4}$$

Thus, for isoprene, the observed concentrations reflect those emitted at distances up to 10 km at typical wind speeds and up to 30 km at extreme wind speeds. Given that the immediate oxidation products of isoprene, methacrolein and methyl-vinyl-ketone, have longer lifetimes and each molecule of either methacrolein or methyl-vinyl-ketone comes from one molecule of isoprene, we combine the three amounts to form a new constituent variable, isoprene plus isoprene oxidation products, Isop+ox, which reflects the contributions of isoprene emissions over distances up to four times greater than isoprene

alone. This variable can also provide other information, since the ratio Isoprene/Isop+ox approaches 1.0 for fresh emissions and 0 for very chemically aged emissions.

**Table 4.** Reported lifetime for the observed BVOCs under OH and $O_3$ oxidation. Obtained from [13,58,59].

| Compound | $\tau_{OH}$ | $\tau_{O_3}$ |
|---|---|---|
| Isoprene | 1.4 h | 1.3 day |
| Methacrolein | 4.1 h | 15 day |
| Methyl-vinyl-ketone | 6.8 h | 3.6 day |
| Benzene | 9.4 day | >4.5 year |
| Toluene | 1.9 day | >4.5 year |
| p-xylene | 6 h | >4.5 year |
| α-pinene | 2.6 h | 4.6 h |
| TMB-1,2,4 | 4.3 h | >4.5 year |
| p-cymene | 1 day | >330 day |
| Eucalyptol | 1.4 day | >110 day |

## 3. Results

Figure 3 presents measurements of targeted anthropogenic compounds and Figure 4 presents an overview of the targeted biogenic compounds measured during COALA-JOEYS. The VOCs are reported as dry air mole fractions in nmol/mol (ppb) but are referred to as "concentrations" throughout the manuscript. The maximum concentrations of BVOCs were observed in summer, while the AVOC concentrations peaked in the colder months.

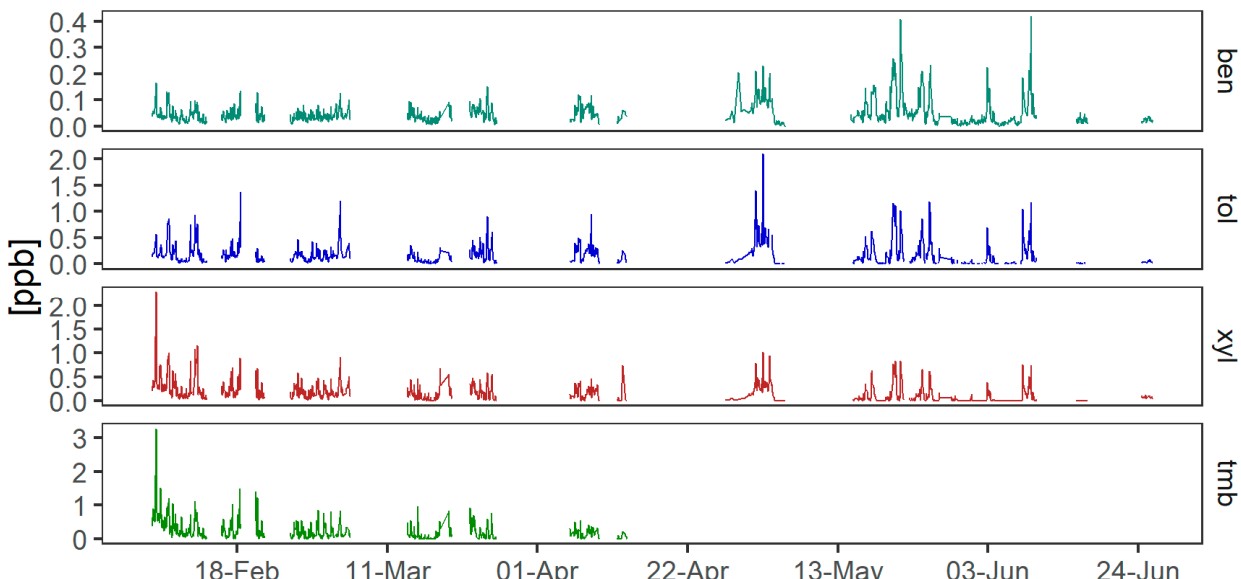

**Figure 3.** Timeseries of AVOCs measured at the ANSTO site using the BAASS. Ben = benzene, xyl = p-xylene, tol = toluene and tmb = 1-2-4-trimethyl-benzene. TMB concentrations were not recorded after mid-April due to an error in the GC-MS analysis method (see Supplementary Materials).

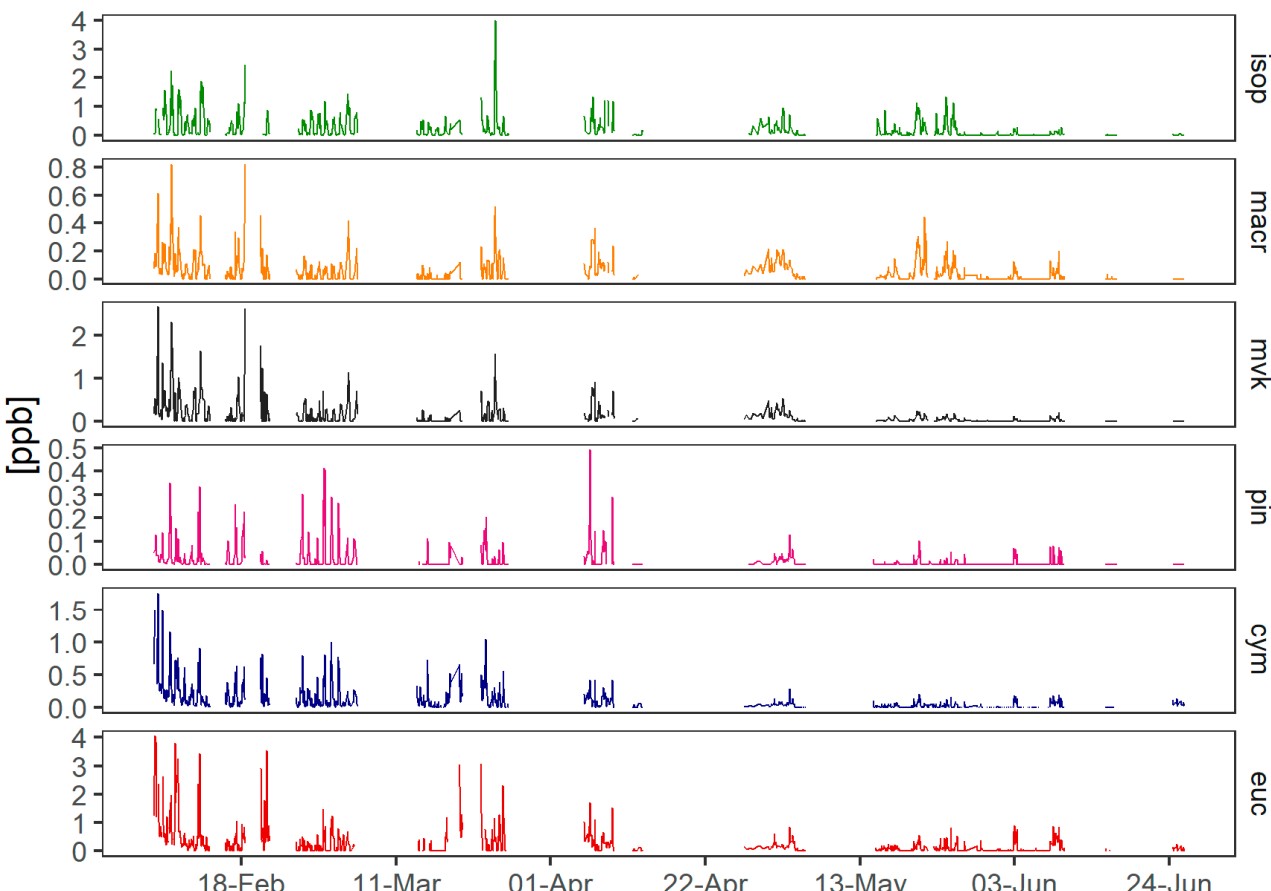

**Figure 4.** Timeseries of BVOCs measured at the ANSTO site using the BAASS, where isop = isoprene, macr = methacrolein, mvk = methyl-vinyl-ketone, pin = α-pinene, cym = p-cymene and euc = eucalyptol. The instrument was malfunctioning during most of April due to contamination in the system.

There is a seasonal change in BVOC concentrations reflected in the high concentrations during summer and in the early autumn days before a decrease when autumn progresses into winter. AVOCs appear to maintain a similar concentration variation during the sampling period but further analysis shows how the mean concentration per month has a decreasing trend (see Discussion).

Figure 5 presents the total daily mean concentration of the six main BVOCs and the three AVOCs measured throughout the campaign. For the VOCs measured, BVOCs constitute the larger fraction during the warmer months, whilst the AVOCs are often the greater fraction of the total in the cooler months. While there are large numbers of other VOCs with much smaller concentrations present in the sampled atmosphere, we expect that the reported totals are likely to represent a major fraction of the total sampled VOCs based on previous studies. Winters et al. [60] analysed the VOC emissions from common *Eucalyptus* species in Australia, finding that most emissions are comprised of isoprene, eucalyptol, p-cymene and a-pinene. AVOCs are a much larger group of species but there is special interest in the carcinogenic aromatic group, including benzene, toluene and xylene [23]. The contribution of aromatics to the overall AVOC depends on the sources impacting the site [46]. Aromatics have a higher contribution close to on-road sources than in suburban environments, with 35% and 16% of total AVOC loading attributed to aromatics, respectively.

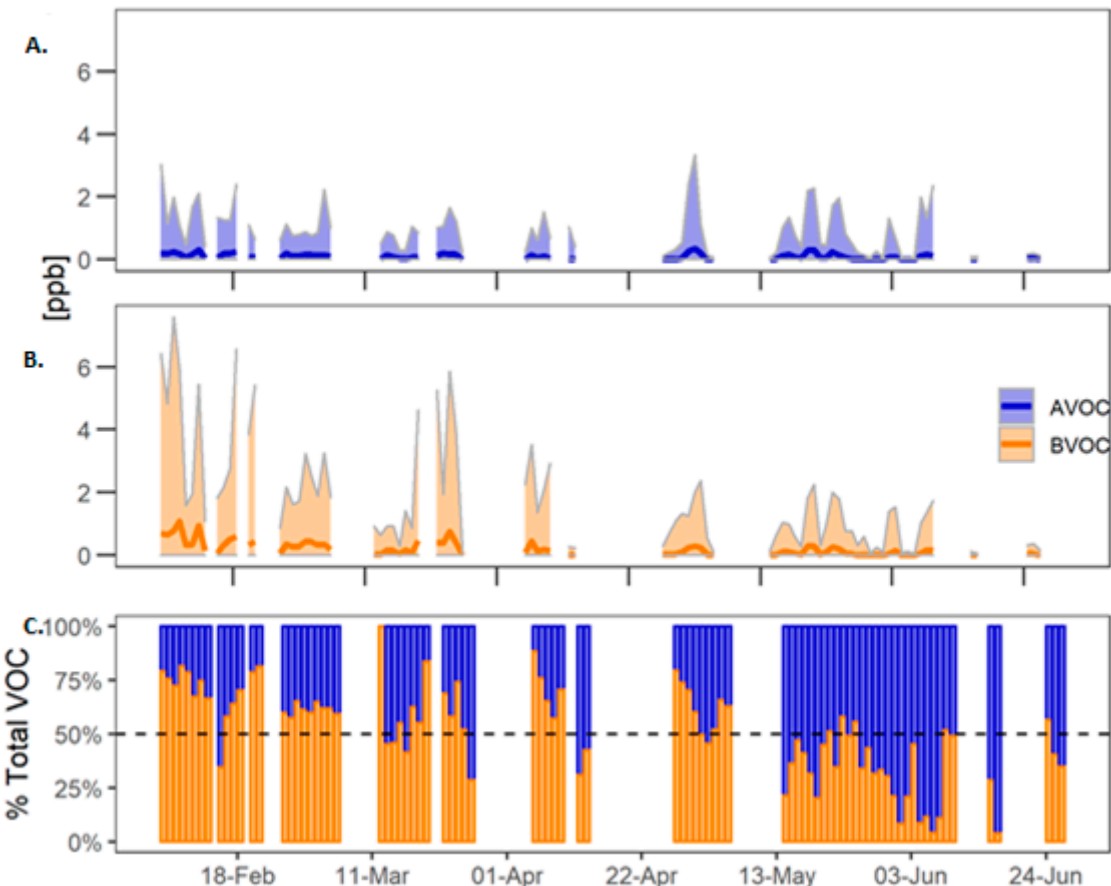

**Figure 5.** Total AVOC and BVOC during COALA-JOEYS. (**A**) Sum of the concentration of benzene, toluene and p-xylene; (**B**) Sum of isoprene, MVK, MACR, eucalyptol, p-cymene and α-pinene. The dark line shows the mean value and the area around the line shows the daily minimum and maximum values. (**C**) Ratio of biogenic and anthropogenic VOCs to the total VOC load excluding TMB.

The BVOCs observed during the COALA-JOEYS campaign exhibit large variability in their ambient concentrations due to their short atmospheric lifetimes and the dependence of the emissions on temperature and sunlight [16]. Peak values of approximately 8 ppb were observed at midday in February, falling to near zero overnight. BVOCs constitute the major fraction of total atmospheric VOCs in summertime and early autumn while AVOCs dominate during the early winter.

## 4. Discussion

### 4.1. Influence of Wind Speed and Direction on VOC Concentrations

During the sampling period, winds coming from the SE and SW were the most frequent (Figure 6A). W-N-NE winds have the higher influence on AVOC concentrations (Figure 6B), with the highest concentrations observed when winds are light from the NW, most likely bringing polluted air from the broader Sydney metropolitan area ~31 km away and the Lucas Heights waste management facility to the west (Figure 1).

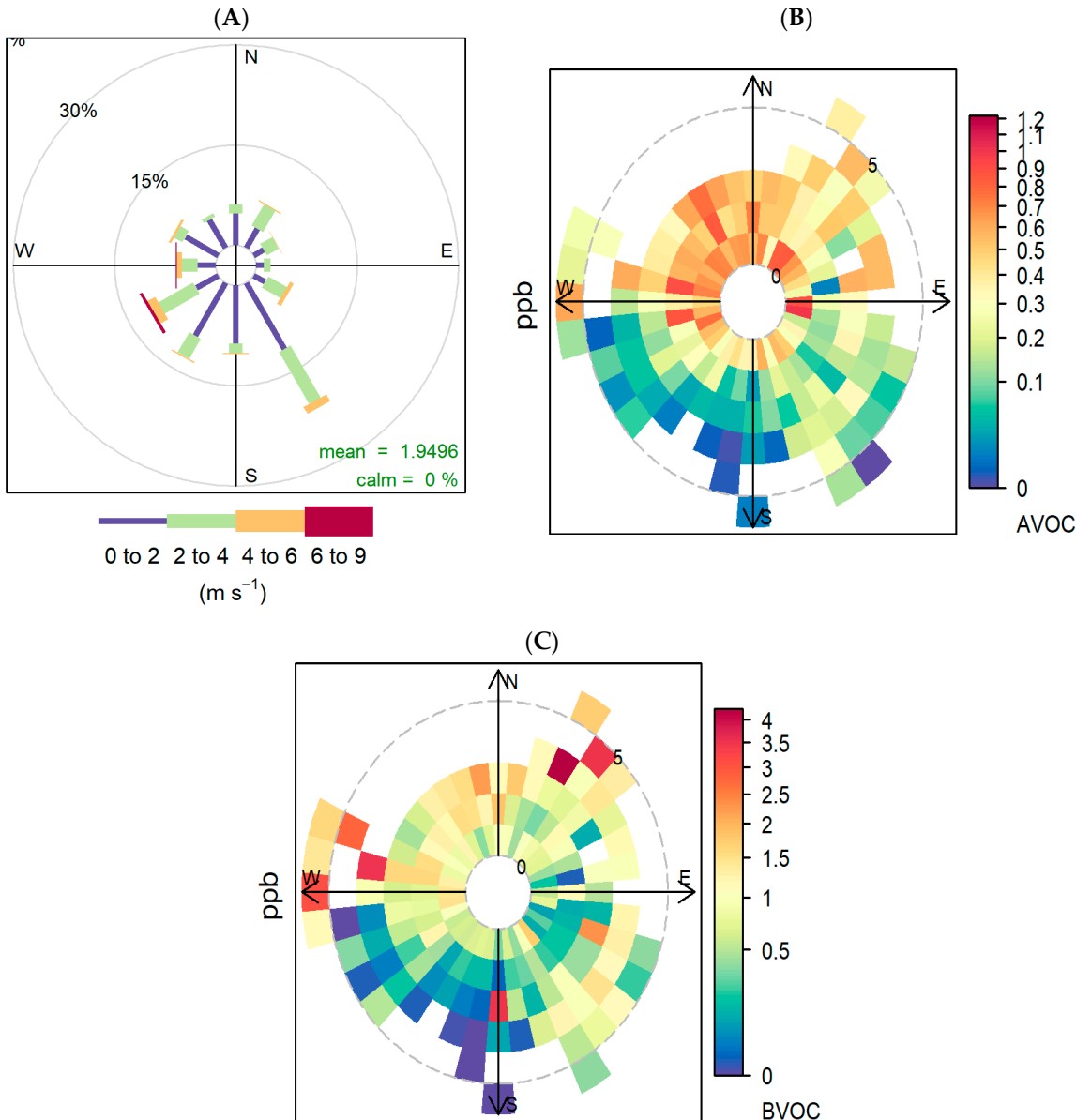

**Figure 6.** (**A**) Wind rose using wind observations at 10-m height for the entire sampling period. Predominance of the SW direction can be observed, with more than 35% of the observations coming from that quadrant, followed by SE winds with 25% of frequency. In addition, 85% of the time, the wind speed was lower than 3 m/s. (**B**) Polar plot using total mean concentrations of targeted AVOCs during COALA-JOEYS. (**C**) Polar plot using total mean concentrations of targeted BVOCs during COALA-JOEYS. Concentrations reported in ppb.

While Figure 6C shows a large variation of BVOCs with wind speed and direction, the month by month analysis shown in Figure 7 indicates that the underlying change in the month by month concentrations of BVOCs is much larger than the concentration differences associated with changes in wind speed and direction. In late autumn and winter (May–June), measured BVOC concentrations are generally low from all wind directions. In contrast, during February (summer), high BVOC concentrations are associated with winds from the west through north to northeast. Winds from the west have an extended fetch over forested regions in that direction, whilst peak BVOC concentrations from the northeast reach the site only when the wind speed is high (ws > 3 m/s) and the temperature peaks (Figure 7). The higher wind speed increases the sampling fetch of the instrument and consequently the mass of oxidised products arriving at the measurement site from the northern forested areas. During the N-NE events, the measured Isoprene/Isop+ox ratio

is typically less than 0.5, indicating relatively aged air, whereas the fresher emissions of isoprene came from the south, with an Isoprene/Isop+ox ratio of around 1 (see Supplementary Figure S5). The forests to the south are closer (<2 km) than the northern forests, giving the isoprene less time to oxidise before reaching the site.

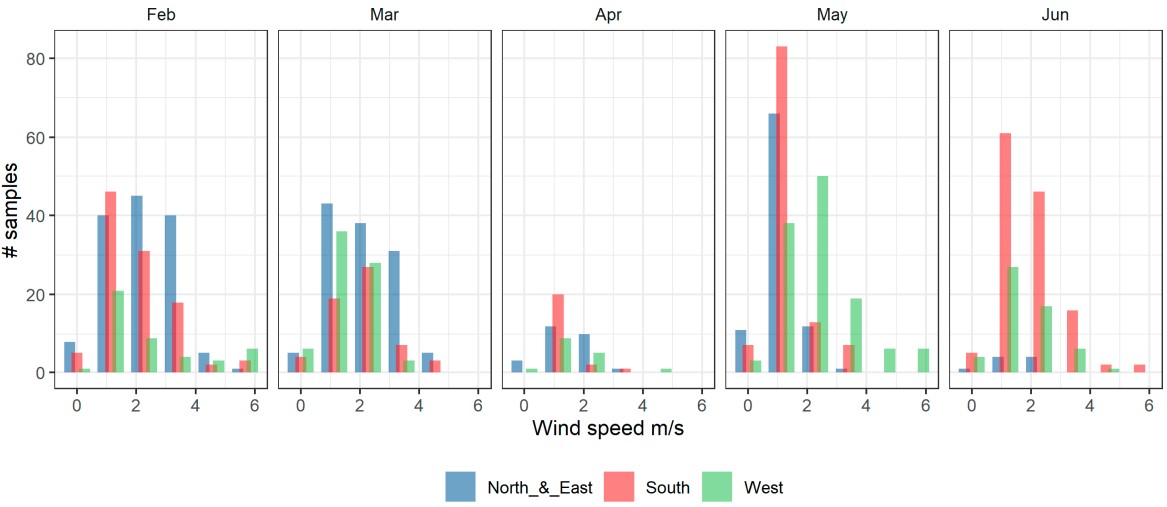

**Figure 7.** Wind speed frequency distribution per month during the COALA-JOEYS. Only the wind speed data of the sampling period were included.

### 4.2. Influence of Temperature and PAR on VOC Concentrations

BVOC and AVOC concentrations both decrease with temperature (Figure 8A,B). The BVOC seasonal trend can be explained by the temperature and light dependence for BVOCs [61–63]. The AVOC emissions, however, depend more on changes in sources close to the site and the influence of meteorology. From February to April, N/NE winds have a greater influence on the sampling site than in May and June (see Supplementary Figure S6). Consequently, the influence of emissions from the Sydney metropolitan area decreased over the period of COALA-JOEYS. Grouping isoprene, MACR and MVK concentrations provides a better understanding of isoprene emissions by taking into account the fact that the sampling location can be influenced by biogenic emissions up to 30 km away, allowing enough time for isoprene reaction. Isoprene has a relatively short lifetime of ~2 h during daytime [13]. MACR and MVK make up ~70% of the products from isoprene oxidation, making them helpful in the analysis of isoprene emissions and the atmospheric oxidation capacity. Dividing BVOCs into two categories, (1) isoprene and its oxidation products (MACR and MVK), and (2) monoterpenes, the temperature–concentration trend is visible in both groups (Figure 8C,D). Both groups have similar concentrations over the summer and early autumn. The increase in isoprene group concentrations observed in April is influenced by the lower number of samples taken and the relatively high influence of northwesterly winds during this period. When the temperature decreases in June, monoterpenes comprise a greater proportion of the total BVOCs, although the total ambient concentration of both is very low. Using only the data with (radon-determined) stable nights results in clearer trends in these results (see Supplementary Figure S7).

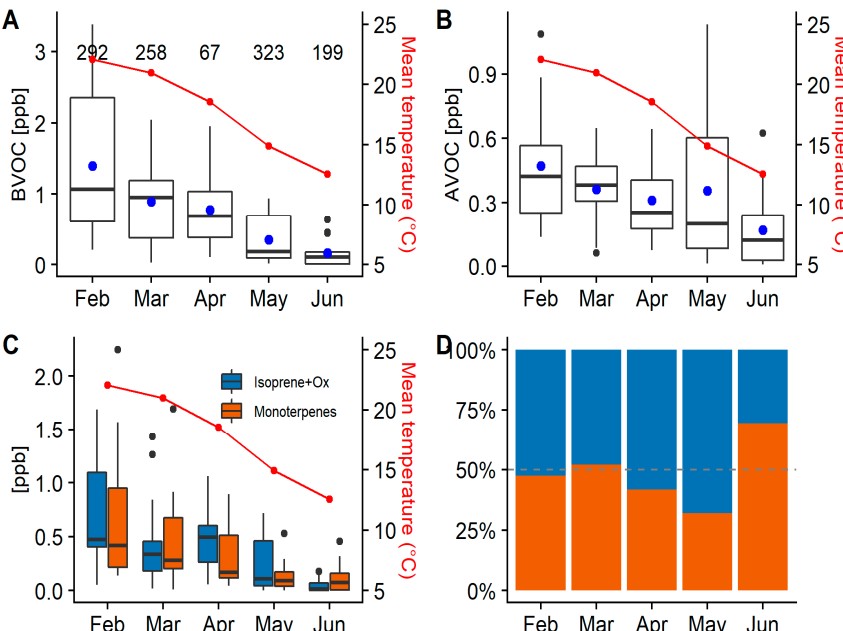

**Figure 8.** (**A**) Box plot showing the variation of the total daily mean targeted BVOC concentration per month, along with the monthly mean temperature (24 h) in red. The black line represents the median, the line on the box is the 95% of the distribution range and outliers are shown as black dots. The number on top of the box plots represents the number of samples per month in the analysis. The blue dot represents the overall mean concentration. (**B**) Box plot showing the total daily mean concentration of targeted AVOCs per month along with the monthly mean temperature in red. Most of the variability observed in May can be explained by periods of high advections and wind speed onsite. (**C**) Daily mean concentration box plot of the monoterpenes (orange) and the sum of isoprene and its oxidation products (blue). (**D**) Relative contribution of BVOC groups (Isop+ox; monoterpenes) estimated from the average concentration of each group divided by the sum of both groups per month.

The dependence on temperature for isoprene presented in Figure 8C is further clarified when analysing a diurnal cycle (Figure 9A,B). The isoprene group follows a clear diurnal pattern where temperature is driving the emissions and hence observed concentrations of isoprene, MACR and MVK. Photosynthetically active radiation (PAR) has a potential impact on isoprene emissions but is a smaller driver than temperature for this dataset (Figure 10). One of the known plant function roles of isoprene is as temperature protection. As long as photosynthesis rates are maintained by incident levels of PAR, temperature will play a larger role in the rate of isoprene emission [61,63,64]. The concentrations of Isop+ox peak around midday and then start decreasing. The decrease could be associated with the continued oxidation of isoprene and its products along with decreased emissions or by the effect of the sea-breeze circulation, decreasing the concentrations at this time [41,42].

The dependence on temperature for monoterpenes presented in Figure 8C is not replicated when analysing a diurnal cycle (Figure 9A,B). During the night, the temperature is cooler and the nocturnal boundary layer will form, decreasing the mixing volume. In the absence of daylight, OH and $O_3$ production will cease, making $NO_3$ the dominant oxidant in the atmosphere. Under these conditions, the atmospheric lifetimes of eucalyptol and p-cymene increase significantly, thereby favouring the accumulation of monoterpenes in the lower nocturnal boundary layer. This is reflected in the greater proportion of monoterpenes in the BVOC loading during the night and early morning. The decrease in monoterpenes during the daytime reflects both the deeper mixing in the planetary boundary layer and the oxidation of monoterpenes by OH. Because of their longer chemical lifetimes, the observed p-cymene and eucalyptol will reflect emissions over a much wider domain than that represented by isoprene and its oxidation products.

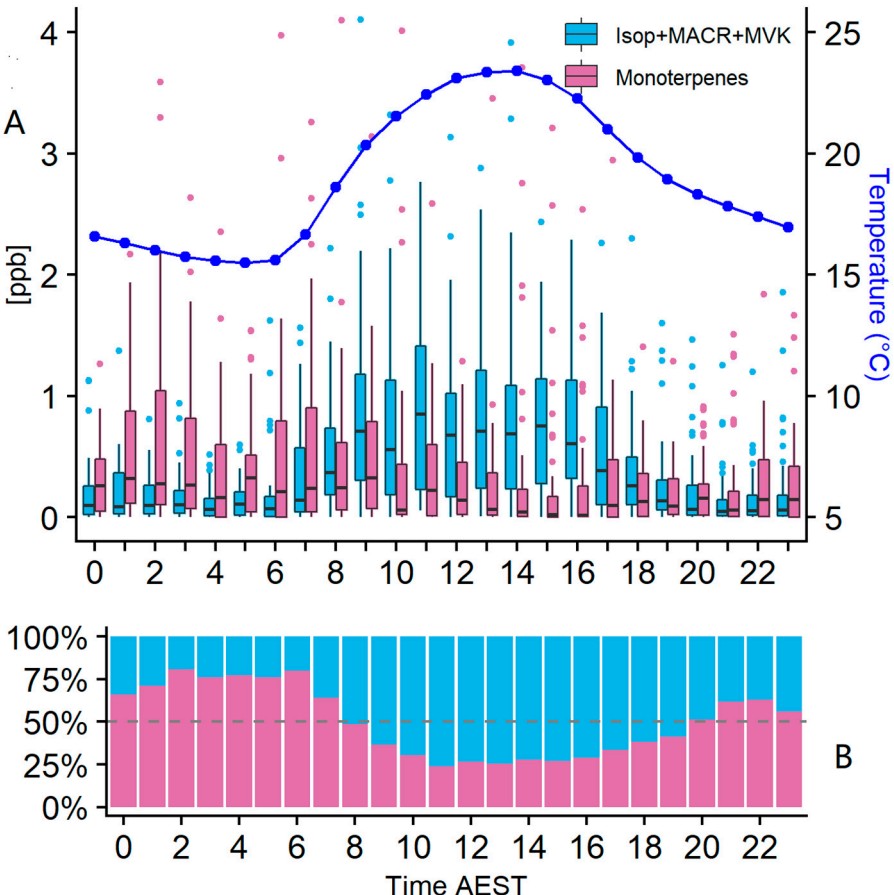

**Figure 9.** (**A**) Composite mean concentration of monoterpenes and isoprene grouped by hour between February and May. On the right y-axis is the average temperature for the analysed period. June data were excluded to capture a better representation of the influence of temperature on VOC concentrations. The black line represents the median, the line on the box is the 95% of the distribution range and outliers are shown as blue and pink. The calculated PAR from the solar radiation observations has a correlation coefficient with temperature of 0.62, and both parameters follow a similar daily pattern influencing the VOC emissions (see Supplementary Figure S8). (**B**) Percentage contribution of BVOC groups to total BVOC loading per hour using mean concentrations.

A multivariate linear regression model was applied to the total isoprene, MACR and MVK measurements, with temperature and PAR as the independent variables. Slopes from the multivariate model showed a greater influence of temperature on isoprene concentrations (slope of 0.14 compared to 0.0002 from PAR) and a $r^2 = 0.51$. To evaluate the independent influence per variable, a nonlinear model was applied to the daylight data of the isoprene group to evaluate the temperature and PAR dependence (Figure 10). When plotting against temperature, an exponential fit is observed (Figure 10A; $r^2 = 0.51$). This result has as large a correlation as the combined temperature and PAR linear model described above, indicating the dominant role of temperature in daylight isoprene levels. The dominant role of temperature has been observed in previous studies measuring emissions from plants and in ambient measurements [20,61,63–65]. The COALA-JOEYS measurements were made in ambient air and not directly from the source; therefore, the effects of other variables influencing isoprene concentration, such as the atmospheric mixing and advection as well as the chemistry happening during transport, are not included in the model. There was no clear relation between PAR and the ambient concentration of isoprene and its oxidation products when using all the daylight data due to very high scatter in the data (see Supplementary Figure S9). There was an especially high variability of isoprene and its oxidation products observed during days with temperatures over 30 °C and winds

coming from the north. These two conditions favoured the emission and transport of isoprene, MACR and MVK from the north and northwest even during periods where no PAR was incident. These isoprene peaks were especially common during the late afternoons where temperature was over 28 °C. Data with temperatures over 20 °C during hours with PAR less than 500 μmol m$^{-2}$s$^{-1}$ were filtered out and then a second filter applied, removing category 1 and periods of nocturnal synoptic non-stationarity (radon stability categories) both characterised by high wind speed and advection. A linear model was applied to the filtered data (Figure 10B). At the COALA-JOEYS site, a combined influence of meteorological and chemistry factors may be masking the influence of PAR on isoprene emissions. A direct measurement of emissions could clarify this influence but that is out of the scope of the present study.

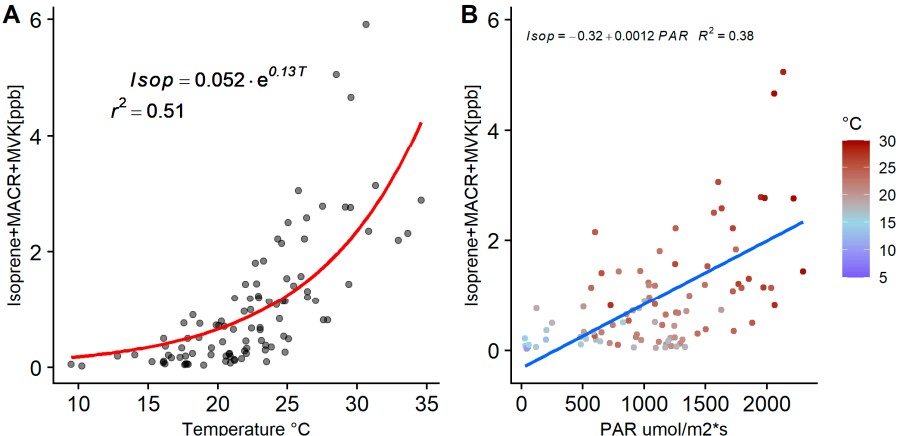

**Figure 10.** (**A**) Nonlinear fit applied to temperature-isoprene correlation plot. Error values were 0.013 and 0.019 for the exponential factor. (**B**) Correlation plot of PAR against isoprene and its oxidation products, coloured by ambient temperature. Isoprene measurements were filtered to exclude transport events. Only the daylight data were used to estimate the linear fit to reduce the noise from the night data, as well as the data coming from SW-N to capture the highest concentrations of Isop-ox in the model.

### 4.3. BVOC Concentrations Observed in COALA-JOEYS Compared to Previous Studies

Table 5 presents the mean BVOC concentrations observed during COALA-JOEYS alongside results from previous studies made in NSW. Measurements during the MUMBA campaign were taken in Wollongong, close to a low-traffic road and a natural reserve, during summer [42]. The SPS1 and SPS2 [66] campaigns were located in the western Sydney urban area during summer and autumn, respectively. MUMBA and both SPS campaigns used the same instrument (PTR-MS) for VOC measurements. All results showed BVOC concentrations on the same order of magnitude except during winter in the present study. To our knowledge, there are no previous measurements of BVOCs in the region during winter. From our measurements, we can see that during winter, BVOC concentrations decrease but they are still present in the atmosphere. Although AVOCs will dominate the atmospheric chemistry given the relatively higher concentrations during these months, BVOCs will still have a role in the local atmospheric chemistry, but at a smaller scale compared to summer. The monoterpene group is the largest contributor to the BVOC loading in the atmosphere during the whole sampling period of the present study. All previous studies reported a lower monoterpene/isoprene ratio. The difference can be attributed to the inclusion of eucalyptol in the monoterpene group in the present study. Eucalyptol is not usually treated as a monoterpene but is a related compound. The ratios between the monoterpene group show that more than 75% of the observed monoterpene mass is eucalyptol, around 20% is p-cymene and 5% is α-pinene. This can be explained by the emission composition and the lifetime of these compounds. Eucalyptol comprises a larger portion of the total BVOC emission from southern Australian vegetation compared

to other monoterpenes [60]. α-pinene has the shortest atmospheric lifetime among the sampled compounds, followed by p-cymene and eucalyptol [13]. Therefore, as eucalyptol is the dominant emitted species of observed monoterpenes and has the longest atmospheric lifetime, it dominates monoterpene measurements in the present study.

**Table 5.** Mean BVOC concentrations in ppb during the COALA-JOEYS 2019 campaign compared to previous studies in Australia. MVK = methyl-vinyl-ketone, MACR = methacrolein, Monoterpenes = sum of p-cymene, eucalyptol and α-pinene. Concentrations are in parts per billion.

| Compound | SPS1 [66] (Summer) | SPS2 [66] (Autumn) | MUMBA [67] (Summer) | This Study | | | |
|---|---|---|---|---|---|---|---|
| | | | | Summer | Autumn | Winter | Overall |
| Isoprene | 1.05 | 0.63 | 0.28 | 0.37 | 0.23 | 0.01 | 0.27 |
| MVK+MACR | 0.78 | 0.36 | 0.2 | 0.32 | 0.16 | 0.00 | 0.21 |
| Monoterpenes | 1.04 | 0.46 | 0.12 | 0.60 | 0.32 | 0.11 | 0.40 |
| Eucalyptol | - | - | - | 0.40 | 0.22 | 0.07 | 0.27 |
| Monoterpenes. non eucalyptol | - | - | - | 0.20 | 0.10 | 0.04 | 0.13 |

Isoprene/Isop+ox ratios were similar during COALA-JOEYS and in the MUMBA campaign observations [67]. This may be explained by similarities between the sampling locations. Both campaign sites were close to forested areas, with low anthropogenic influence. The measurements during SPS1 and SPS2 show the same seasonal trends as observed during this study. Isoprene, MACR, MVK and monoterpene concentrations decrease with temperature in both instances. However, the magnitude of observed concentrations is a factor of between 2 and 3 higher in the SPS results. A variety of factors could be driving this difference—for example, the sampling location or the analysis method and instrument in each campaign [68]. The isoprene/Isop+ox ratio close to 0.5 indicates an oxidised isoprene measurement during most of the time for MUMBA and this study, but SPS1 and SPS2 reported fresher isoprene observations with ratios >0.6.

### 4.4. Understanding the Variation in Observed AVOC Concentrations and Ratios

Table 6 compares mean AVOC concentrations from COALA-JOEYS with Australian measurements from other sampling environments. A canister deployment in the Clem Jones Tunnel, Brisbane, which links two major roads, is referred to as the "tunnel" study. The samples were analysed using GC-MS [47]. The same technique was used to sample on a major road in Brisbane ("on-road") as well as 500 m from a major roadside ("suburban") [46]. The "outdoor" study reports concentrations from locations outside of multiple dwellings in southeastern suburban Melbourne. The samples were collected during different seasons in 2008/9 and were analysed using GC-FID-MS [25]. Our measurements showed low AVOC concentrations when compared to most of these other environments (Table 6), suggesting that anthropogenic sources do not have a large influence at our sampling site. This is similar to the observations during the MUMBA campaign, where mean AVOC concentrations were of comparable magnitude to the current study.

Comparing the February to April period to the May to June period, the concentration of benzene increased slightly, toluene remained stable, but p-xylene concentration was reduced as the year progressed (see Table 6). Dividing the data by wind direction shows that northern, eastern and southern winds show similar trends where benzene, toluene and xylene concentrations increase until May (as atmospheric mixing decreases) and then decrease by up to a factor of 4 in June (see Figure 11). In contrast, the winds coming from the west show a different pattern, with toluene and xylene decreasing as the year progresses into winter and benzene staying relatively stable and then increasing in June.

**Table 6.** Mean AVOC concentrations during the COALA-JOEYS 2019 campaign compared to previous studies in Australia. All concentrations are reported in ppb. TMB 1.2.4 = Trimethyl-benzene 1,2,4. The two columns represent the period where TMB 1.2.4 signal was recorded (Feb–mid-April) and the period without TMB measurements (mid-April–June).

| Compound | Tunnel [47] | On-Road [46] | Suburban [46] | Outdoor [25] | SPS1 [66] | SPS2 [66] | MUMBA [67] | This Study Feb–April | This Study May–Jun |
|---|---|---|---|---|---|---|---|---|---|
| Benzene | 4.4 | 1.3 | 0.5 | 0.24 | 0.3 | 0.6 | 0.11 | 0.04 | 0.05 |
| Toluene | 9.3 | 4.9 | 2.5 | 1.01 | 0.8 | 2.18 | 0.3 | 0.17 | 0.17 |
| p-xylene | 6.6 * | 1.7 * | 1.3 * | 0.32 | 0.53 | 1.52 | 0.3 | 0.19 | 0.09 |
| TMB-1,2,4 | 2 | 0.4 | 0.5 | 0.14 | 0.37 ** | 0.82 ** | 0.15 ** | 0.22 | - |

* Reported as the summed concentration of o-xylene and p-xylene. ** Reported as the summed concentration of TMB isomers.

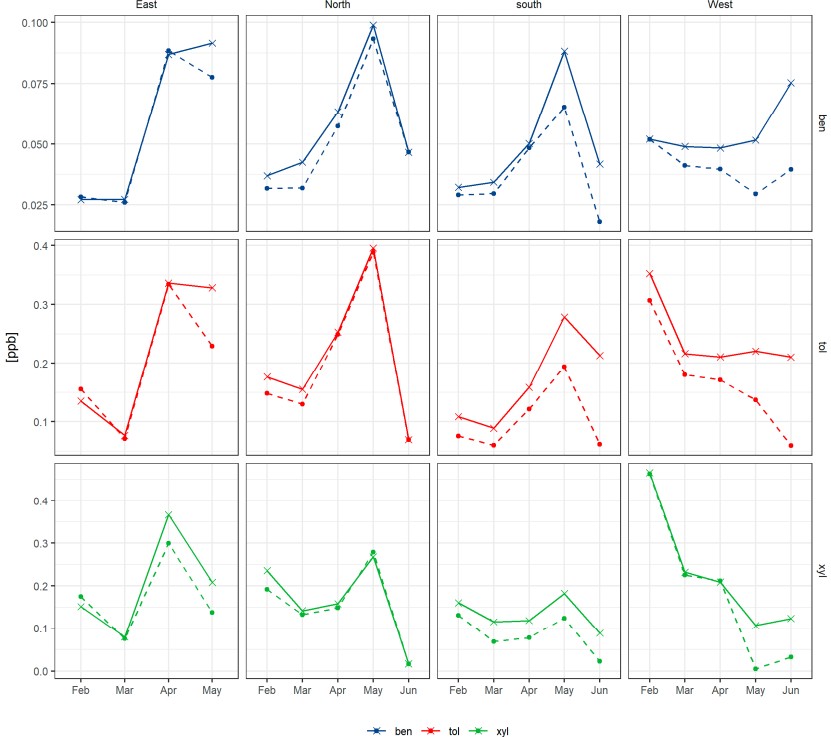

**Figure 11.** Mean concentration (solid line) and median concentration (dashed line) for benzene, toluene and xylene each month during COALA-JOEYS using only the night data with stable conditions. The columns represent the different sectors from the site and the rows are "ben" = benzene; "tol" = toluene and "xyl" = xylene.

Comparing the observed trend in AVOC concentrations by months with westerly winds to the monthly emissions from the NSW emission inventory for the suburbs to the west of the site, a similar behaviour is observed (see Figure 12). The decrease in toluene and xylene is attributed mainly to the decrease in mowing activity in the area, while the increase in benzene is attributed to the increase in residential wood heating emissions, which are estimated to increase up to a factor of 30. The concentration increase from other sources can be explained by the effect of reduced mixing caused by the increase in the frequency of low wind speeds coming from these directions, favouring accumulation during the night (see Supplementary Figure S10). The decrease in AVOCs observed in the other directions in June is explained by the low influence of northern and eastern winds onsite during the sampling time in June as well as the decrease in the temperature diminishing the evaporative emissions from most of the sources.

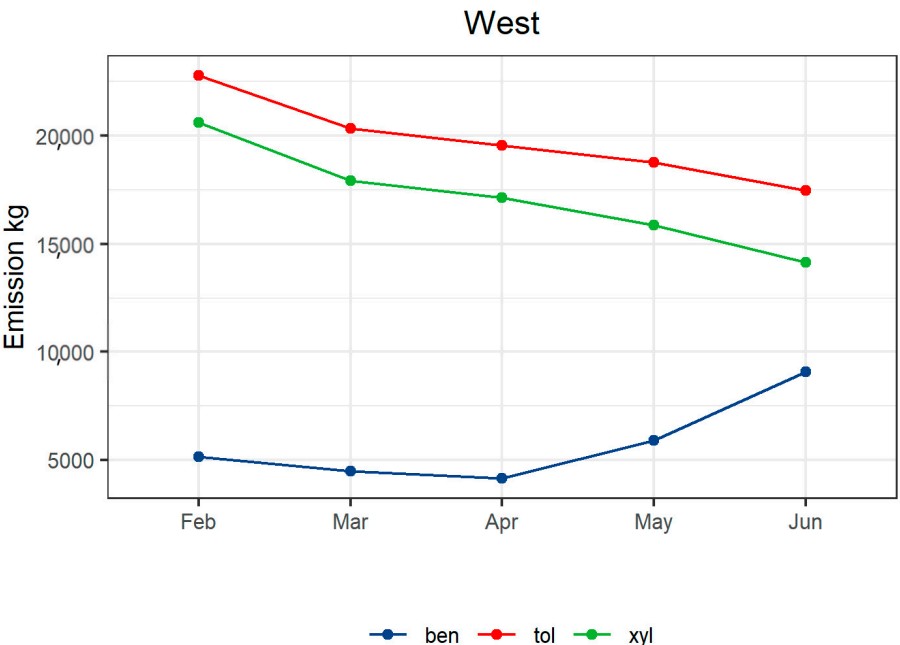

**Figure 12.** Monthly emissions from the NSW emission inventory from the suburbs located west of the sampling site. Ben: benzene, tol: toluene and xyl: xylene.

The semi-rural location of our sampling site for COALA-JOEYS explains the relatively low concentrations that we observed relative to previous studies and further information about sources can be obtained by considering the ratios of the measured AVOCs. The xylene/toluene and benzene/toluene ratios observed during COALA-JOEYS are similar to previous studies (Table 7). The main difference is the xylene/toluene ratio, which changes between studies. This can be explained by the shorter relative lifetime of p-xylene and the influence of local sources. The tunnel and on-road measurements from previous studies were closer to the AVOC source [46,47] and the direct influence of nearby on-road traffic sources generates higher concentrations of BTEX at these locations. In contrast, at the COALA-JOEYS site, average concentrations of benzene and toluene were below concentrations reported at suburban sites [25,46], meaning relatively cleaner air with some influence of on-road traffic sources reaching the site.

**Table 7.** Mean AVOC ratios normalised to toluene.

| Ratio to Toluene | Tunnel | On-Road | Suburban | Outdoor | SPS1 | SPS2 | MUMBA | This Study | |
| | | | | | | | | Feb–April | May–Jun |
|---|---|---|---|---|---|---|---|---|---|
| Benzene | 0.47 | 0.27 | 0.20 | 0.24 | 0.38 | 0.28 | 0.37 | 0.24 | 0.33 |
| p-xylene | 0.71 | 0.35 | 0.52 | 0.32 | 0.66 | 0.70 | 1.00 | 1.12 | 0.62 |
| TMB-1,2,4 | 0.22 | 0.08 | 0.20 | 0.14 | 0.46 | 0.38 | 0.50 | 1.24 | - |

A significant difference when comparing observed VOC concentrations in these different environments is the ratio of TMB-1,2,4 to other AVOCs. The observed concentration of TMB-1,2,4 in this study is similar to an on-road or suburban environment but the TMB/toluene ratio is larger than values previously reported. This could be due to the AVOC emission ratios of local sources such as coal mining, fuel storage or the municipal dump close to the area; however, the exact cause is not known.

Using the NSW emission inventory, emissions within a radius of 50 km from the sampling site were used to estimate benzene/toluene and xylene/toluene ratios for domestic sources, the 15 highest emitters and an overall ratio for all sources, presented in Table 8.

**Table 8.** Estimated ratios from the NSW emission inventory 2013.

| Ratio | Benzene/Toluene | Xylene/Toluene | Benzene/Xylene |
|---|---|---|---|
| Overall | 0.25 | 0.8 | 0.31 |
| Highest emitters | 0.39 | 0.66 | 0.59 |
| Domestic | 0.21 | 0.91 | 0.26 |
| This study: Feb–April | 0.24 | 1.12 | 0.21 |
| This study: May–June | 0.33 | 0.62 | 0.53 |

The data were divided by wind direction into three directions: north and east (bringing influences from Sydney), west (inland suburbs) and south (Wollongong and Pork Kembla). Only the night data with moderately stable and mostly stable conditions were used for the analysis. The analysis shows different trends for all directions (see Figure 13). The north and east sector maintains a benzene to toluene ratio with low variation from February to May, varying between 0.20 and 0.26. A much higher benzene to toluene ratio is observed during June but coincides with a very low influence of north and east winds during this month, making it inconclusive. The south region shows an increase in the benzene/toluene ratios from February to May. The southerlies increase their influence onsite, bringing more air from high benzene emitters at the steelworks at Port Kembla. The highest benzene to toluene ratios were observed coming from this sector, a clear influence of the steelworks emissions onsite, as shown in Figure 14. The slight decrease in June is explained by the lower influence of southern winds during the sampling time in June, when western winds were more frequent. The western winds show a continuous increase in the benzene to toluene ratio along the sampling period, agreeing with the NSW emission inventory monthly variation. Although the benzene/toluene ratios are in the same range in the south and the west, the west showed higher concentrations of benzene and toluene, probably influenced by the distance to the source. The proximity of the Lucas Heights landfill (~1 km) and Campbelltown area (~14 km) compared to Port Kembla and Wollongong (~50 km) allows for less dilution of the plume impacting the site.

Evaluating the whole sampling period, all sectors have a benzene/toluene ratio similar to the overall NSW emission inventory ratio. The xylene/toluene ratios show a similar trend in all directions, shifting from high ratios during February (>1) to lower ratios in June (0.6–0.7). The decrease might be influenced by the decreasing trend in the evaporative emissions and exhaust emissions from mowing and recreational boats (highest emitters in the area) as well as the seasonal changes in meteorology. The change in both ratios reflects a change in how local emissions and regional emissions are impacting the site. During the summer months, with higher emissions and higher advection, the ratios show agreement with the overall NSW emission inventory ratio, with the highest xylene/toluene ratios and the lowest benzene/toluene ratios. When the season changes and atmospheric mixing decreases, the local emissions seem to play a more important role, with higher benzene/toluene ratios and low xylene/toluene ratios.

*4.5. Policy Implications*

Significant BVOC emissions in Southeast Australia dominate anthropogenic VOC emissions during periods of hot weather. Ozone pollution events occur in populated regions of Southeast Australia overwhelmingly during hot days—days on which BVOC emissions from urban and forest vegetation would be expected to be high. It can therefore be deduced that policy designed to reduce anthropogenic VOC emissions with the goal of preventing ozone pollution events may be ineffective owing to the likely dominance of BVOCs in the atmosphere on hot days. (Note that the authors encourage policy reducing AVOC emissions regardless, as these species are toxic—simply that BVOCs must also be considered). To reduce urban BVOC emissions, emissions on a species level should be

considered when selecting vegetation for urban greening projects. This will likely require further research in the Australian context.

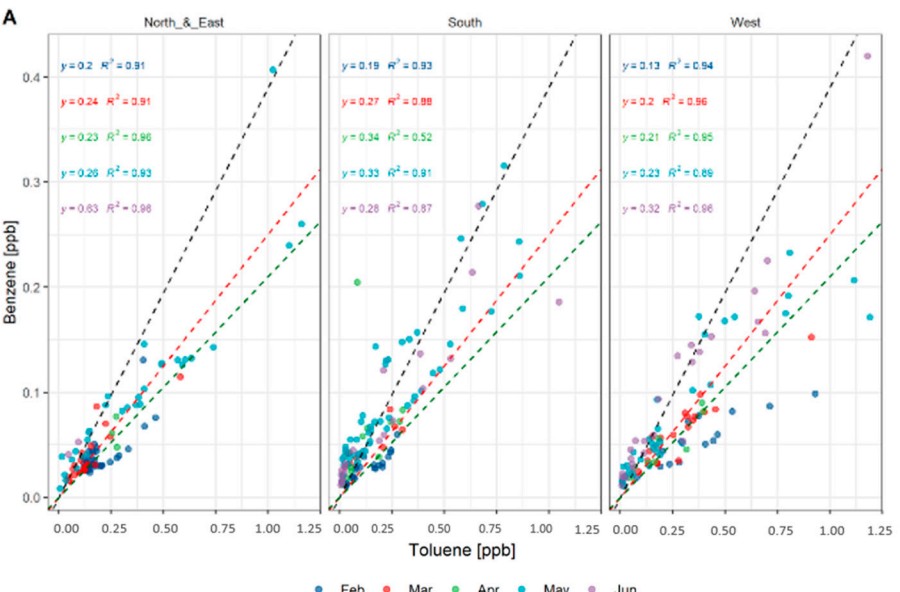

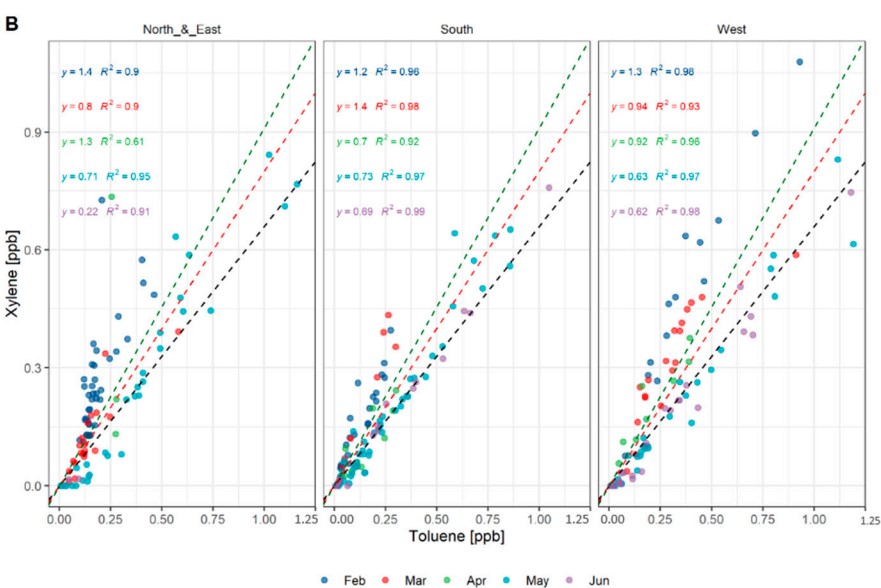

**Figure 13.** (**A**) Correlation plots of benzene to toluene, comparing our observations during the whole sampling period of COALA-JOEYS to the NSW emission inventory ratios. The black dotted line represents the NSW emission inventory ratio for the highest emitters (0.39). The red dotted lined represents the ratio for all the emissions in the area (0.25). The green dotted line represents the domestic + commercial emissions ratio (0.21). The value in the linear model represents the benzene/toluene ratio estimated for each month using a linear model. The colour of the points represents the month. Models for April might not be representative given the low number of samples taken during this month. (**B**) Correlation plots of xylene to toluene, comparing our observations during the whole sampling period of COALA-JOEYS to the NSW emission inventory ratios. The black dotted line represents the NSW emission inventory ratio for the highest emitters (0.66). The red dotted lined represents the ratio for all the emissions in the area (0.8). The green dotted line represents the domestic + commercial emissions ratio (0.91).

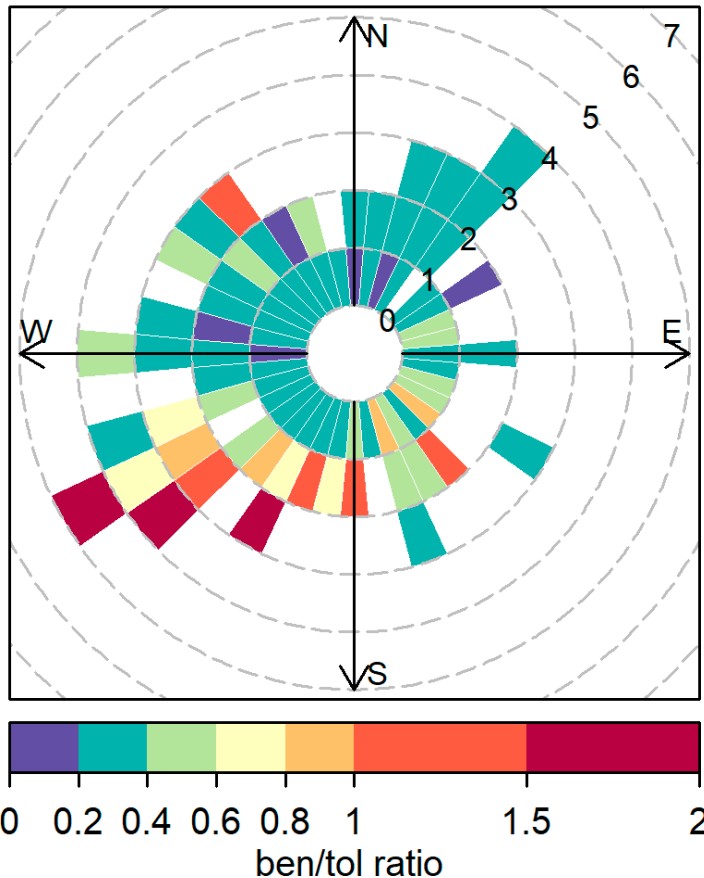

**Figure 14.** Polar plot showing the benzene to toluene ratios during nighttime and stable atmosphere.

## 5. Conclusions

The COALA-JOEYS campaign has provided measurements of ten VOCs during three different seasons in 2019 in an area influenced by forest and on-road traffic emissions. The changes in environmental conditions during the seasons influenced the biogenic and anthropogenic VOCs, with higher concentrations observed during hotter days. This difference in atmospheric composition is therefore an important consideration when formulating and evaluating new air pollution control policies over cities like Sydney, where there is an important biogenic influence during the summer and autumn seasons.

Biogenic VOCs were dominated by isoprene, methacrolein and methyl-vinyl-ketone during the summer and the hotter autumn days, while monoterpenes dominate during winter. The relatively high concentration of monoterpenes during winter could have an important role for atmospheric chemistry over Southeastern Australia. Observed AVOC concentrations were similar to locations with low influence from on-road traffic sources. The measured species comprise most of the locally emitted BVOC species and an important part of AVOCs.

The AVOC ratios showed the influence of different sources during the sampling period. The wind direction and wind speed change the sources influencing the site during the day. The site was often influenced by emissions transported from the nearest urban sites and occasionally by farther sources. The observations showed agreement with the NSW emission inventory overall emissions and with domestic + commercial emissions ratios for the early months of the campaign. During the colder months, the observed ratios were higher than the NSW emissions inventory estimates due to the increased influence of local sources.

**Supplementary Materials:** The following are available online at https://www.mdpi.com/2073-443 3/12/1/47/s1, Figure S1A: Instrument inlet along with a secondary meteorological station located at the roof of ANSTO facilities. Figure S1B: Picture of the system including the different cylinders used during the campaign. Figure S1C: General diagram of the system going from the inlet to the Mass Spectrometer. Figure S2: Chromatogram comparing a dry calibration sample (blue) to a calibration sample diluted with moist ambient air. The peaks in order from top to bottom and left to right are: 1. isoprene, 2. methacrolein, 3. methyl-vinyl-ketone, 4. benzene, 5. toluene, 6. p-xylene, 7. a-pinene, 8. 1-2-4 trimethylbenzene 9. p-cymene, 10. eucalyptol. The difference in peak height is caused by the difference in mass used for the samples when using the calibration gas and the diluted mix. Figure S3: Result of the linearity test of the BAASS system to multiple volumes of analyte. Where isop=isoprene, macr= methacrolein, mvk= methyl vinyl ketone, cym= p-cymene, pin= $\alpha$-pinene, ben= benzene, xyl= p-xylene and tol= toluene. Figure S4: Polar plot using mean concentrations of the sum of BVOCs during the sampling period separated by month. Concentrations are reported in ppb. Figure S5: Isoprene to isoprene + MACR + MVK ratios. Left plot is day-time data, right plot is night-time data. The black line represents a ratio of 1, representing fresh emissions. The red dotted line represents a ratio of 0.25, where the total concentrations are mostly oxidated products. There is a clear trend showing the fresher emissions coming from the south (yellow) during day time. There are higher total concentrations (Isop+ox) coming from the north (purple) but a considerable part of this mass is oxidized isoprene (lower isop-ratio). Figure S6: Wind roses by month during the first half of 2019 at the sampling site. The N-NE components influencing the site decrease with each passing month, hence the influence of metropolitan Sydney emissions over the site decreases from summer to winter. Figure S7: Data filtered to keep the stable nights to reduce variability caused by changes in meteorology. Only the moderate stable and mostly stable data (as determined by the radon analysis) was used for this analysis. Figure S7A: Box plot showing the variation of the total daily mean targeted BVOC concentration per month, along with the mean temperature of the sampling period (24 hrs) in red. The black line represents the median, the line on the box is the 95% of the distribution range and outliers are shown as black dots. The number on top of the box plots represents the number of samples per month in the analysis. The blue dot represents the overall mean concentration Figure S7B: Box plot showing the total daily mean concentration of targeted AVOCs per month along with the monthly mean temperature in red. Figure S7C: Daily mean concentration box plot of the monoterpenes (orange) and the sum of isoprene and its oxidation products (blue). Figure S7D: Relative contribution of BVOC groups estimated from the average concentration of each group divided by the sum of both groups per month. Figure S8: Composite mean concentration of monoterpenes and isoprene groups by hour between February and May. On the right y-axis is the hourly average PAR for the analysed period. Supplementary Figure S9: The concentration of the sum of isoprene and its oxidation products plotted against PAR. Data points are coloured using temperature. The high scatter in the data can be explained by the influence of days with temperatures over 30 °C. During these days the isoprene emission and the MACR and MVK formation is higher compared to days with lower temperatures, promoting transport of isoprene and its oxidation products. Figure S10: Wind speed frequency distribution per month during the COALA-JOEYS divided in day and night time. Only the wind speed data of the sampling period was included. Table S1: Composition of the calibration cylinder used during the sampling campaign. Cylinder number CC511971 manufactured by Air Liquide, Houston, TX, USA analysed 06/12/2018. Table S2: Carryover percentage to calibration. Total number of calibrations during the sampling period and results from the data filtering applied to the calibration record per compound.

**Author Contributions:** Conceptualization, J.R.-G.; Data curation, J.R.-G., I.G., A.D.G. and S.D.C.; Formal analysis, J.R.-G., C.P.-W. and I.G.; Funding acquisition, C.P.-W. and A.G.W.; Investigation, J.R.-G.; Methodology, J.R.-G., C.P.-W., I.G., J.S., E.-A.G. and S.D.C.; Project administration, C.P.-W. and A.G.W.; Supervision, C.P.-W. and I.G.; J.R.-G.; Writing—original draft, J.R.-G. and I.G.; Writing—review & editing, J.R.-G., C.P.-W., I.G., J.S., E.-A.G., A.D.G., S.D.C. and A.G.W. All authors have read and agreed to the published version of the manuscript.

**Funding:** This research was funded by Australia's National Environmental Science Program through the Clean Air and Urban Landscapes hub.

**Institutional Review Board Statement:** Not applicable.

**Informed Consent Statement:** Not applicable.

**Data Availability Statement:** Not applicable.

**Acknowledgments:** We thank Leisa Dyer and other staff from ANSTO for providing the lab space to deploy the instruments, the radon and the meteorological data used in the analysis. We also thank them for their time, kindness, support and accommodation during the COALA-JOEYS campaign. We also thank David Griffith and Graham Kettlewell for the loan and deployment of the FTS instrument.

**Conflicts of Interest:** The authors declare no conflict of interest.

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
