# Peer review of "Seasonal Variation of Biogenic and Anthropogenic VOCs in a Semi-Urban Area Near Sydney, Australia"

_atmosphere, doi:10.3390/atmos12010047_

Round 1

Reviewer 1 Report

Seasonal variation of biogenic and anthropogenic VOCs in a semi-urban area near Sydney Australia

Overall Comments

The paper by Ramirez-Gambo et al., involves a study on biogenic volatile organic compounds (BOV) and anthropogenic volatile organic compounds (AVOC) mixing ratios near Sydney area. In these studies, the effect of seasonal environmental conditions such as solar radiation, temperature, photosynthetically active radiation (PAR), and wind speed on the atmospheric BOV and AVOC mixing ratios are investigated. Overall, the paper is well-written and Figures are well-prepared. The work is very suitable for publication in the Atmosphere journal.

My main concern in this paper is that the data presented is only for short periods of time and only for one year or cycle. To have a more convincing and thorough picture of the atmospheric mixing ratios of the associated VOCs in the region of interest, I encourage the authors to obtain data for at least two cycles (or more). I also note that the authors did measurements of CO, CH4, CO2 but there is no further mention of how the measurements involving these gases were utilized in obtaining further chemistry insights on the role they play on the atmospheric fate of the VOCs. To follow up on this, at a minimum, the authors should provide basic calculations or a simple model to show whether CH4 and CO mixing ratios have any seasonal pattern that can be traced or correlated to the oxidation reactions involving the VOCS. To do this, the authors need to evaluate more precisely the effect the OH reactions with the VOCs to see if there is any effect on the CO and CH4 budget. That will give deeper insights on the seasonal patterns presented in this study. Otherwise, there is no point of mentioning that these gas mixing ratio measurements were taken if the authors don’t discuss their importance later in the manuscript.

Specific Comments.

  • The manuscript contains an error message ‘Error! Ref. Source not found.’ This need to be fixed.
  • In the Sampling and Analysis section, the authors have not mentioned how they made quality control measures to eliminated point sources associated with building vents or other point sources from the surroundings
  • In Fig. 2, please state, in the figure, which color represent hot, cold, and warm
  • In Fig 2 (a) and (b), what are the units associated with the x-axis?
  • In section 2.8, is there any chemistry associated with isoprene and the other VOCs in the course of travel? What would affect the change in the mixing ratios during the horizontal movement.
  • The data in Figure 3 and 4 is only for one cycle. Is it possible to incorporate studies involving more than one cycle (See my general overall comments)? This will be helpful since one can see whether there is any trend.
  • From Fig. 4, we see a huge spike in isoprene mixing ratios that goes close to ~4 ppb. Why is this? Is this spike in isoprene levels expected to persist in all cycles? I think this is where another year of study would be beneficial to get two data points (at least).
  • Fig 5. Please provide labels A, B, and C inside the graph.
  • The data in Figure 5 (c)would be represented more clearly in form of a Table.
  • Line 365-369.The authors mention that compounds with an aromatic group have a higher contribution close to on-road sources that in the suburban environments. In the discussion, I expected the authors to provide a more detailed discussion on this point, but I did not see further explanations for this observation.

Author Response

We thank the referees for their time spent carefully reviewing the manuscript, and in their opinions regarding the science and presentation of the material. In the attached file the referee’s comments are in black and the authors responses are in red.

Reviewer 2 Report

The paper by Gamboa et al. entitled “Seasonal variation of biogenic and anthropogenic VOCs in a semi-urban area near Sydney Autralia” is a study on seasonal variation of VOCs with the aim of formulating air pollution control policy. The authors examined the seasonal variation of anthropogenic and biogenic VOCs near Sydney Australia where few VOC measurements have been available. They investigated 10 selected VOCs based on GC-MS measurements between February and June of 2019 in a semi-urban area between natural eucalypt forest and the Sydney metropolitan fringe. They found that BVOC concentrations reflect seasonal changes in environmental conditions, with summer peak dropping to the lowest values in winter. While isoprene, MACR, MVK dominated during summer and early autumn, terpenes dominated by eucalyptol got the larger fraction during winter. They also reported that AVOC concentrations can be explained by the seasonal changes in meteorology and the emissions in the area and reproducible by the NSW emissions inventory. They concluded that the seasonal variations of BVOCs is an important variable for formulating air pollution control policies over Sydney given their source strength during summer, autumn and winter.

The methods, data and discussion are appropriate to make the conclusion. It will be interest to readers involved in studying air pollution and who relating to formulating air pollution policies.

Minor comment:

Many parts of reference are missing at Lines 151 205 231263 273, 282, 296 318, 334, 335 357 376 378 380 390 391 406 407 418 b438 439 443 449 474 475 451 484 540 559 562 567587 615 622, and possiblly other lines.

Table 4 caption, check “??”

Author Response

Response to Reviewer 2 Comments

We thank the referees for their time spent carefully reviewing the manuscript, and in their opinions regarding the science and presentation of the material. In what follows the referee’s comments are in black and the authors responses are in red.

Point 1: Many parts of reference are missing at Lines 151 205 231263 273, 282, 296 318, 334, 335 357 376 378 380 390 391 406 407 418 b438 439 443 449 474 475 451 484 540 559 562 567587 615 622, and possiblly other lines.

Response 2: We thank the reviewer for pointing this out. We have corrected all the missing references in the documents.

Point 2: Table 4 caption, check “??”

Response 2: We have corrected the table caption.

Reviewer 3 Report

In this paper, authors decided to investigate seasonal variations in VOCs concentration in Australia. Introduction describes well previous studies on the matter giving insight for a foreigner to the local Australian situation with VOCs production. Some insight in the project of which this research is part is also given in introduction and results are well presented and explained.

Therefore, I recommend publishing this work after making some corrections listed below:
- in lines 151, 205, 231, 263, 273, 282, 296, 299, 318, 334, 335, 357, 376, 378, 380, 390, 391, 406, 418, 438, 439, 443, 449, 450, 474, 475, 484, 492, 540, 559, 562, 567, 587, 615, 622 there are some problems with references probably caused by the software used (Mendeley, Zotero, Endnote or similar)
- there are also some parts of the text written in red colour (for example in lines 269 and 270)
- on figure 2a) and 2b), apscise is not titled correctly. I presume that authors meant 35th, 40th and 45th week of the 2019, but it is not 100% sure. Also, I did not find in the text which concentration of Radon is considered normal which I would prefer to see somewhere in the text.

Generally, I would prefer that authors use IUPAP and IUPAC recommendations for writing units and quantities, for example: c(compound)/mol dm-3 or φ/ppb or t/°C etc.

Author Response

Response to Reviewer 3 Comments

We thank the referees for their time spent carefully reviewing the manuscript, and in their opinions regarding the science and presentation of the material. In what follows the referee’s comments are in black and the authors responses are in red.

Point 1: in lines 151, 205, 231, 263, 273, 282, 296, 299, 318, 334, 335, 357, 376, 378, 380, 390, 391, 406, 418, 438, 439, 443, 449, 450, 474, 475, 484, 492, 540, 559, 562, 567, 587, 615, 622 there are some problems with references probably caused by the software used (Mendeley, Zotero, Endnote or similar)

Response 1: We thank the reviewer for pointing this out. We have corrected all the missing references in the documents.

Point 2: there are also some parts of the text written in red colour (for example in lines 269 and 270)

Response 2: We have fixed the error

Point 3: on figure 2a) and 2b), apscise is not titled correctly. I presume that authors meant 35th, 40th and 45th week of the 2019, but it is not 100% sure. Also, I did not find in the text which concentration of Radon is considered normal which I would prefer to see somewhere in the text.

Response 3: We apologize for the confusion caused by the apscize on figure 2. We have changed it to decimal days of the year.

Regarding the second part of the comment, there is no “normal” concentration of Radon. As stated in lines 290-293

“Since 222Rn is a naturally-occurring, noble, radioactive gas with a horizontally-distributed surface-only source, changes in its concentration on short (diurnal) timescales are a useful proxy for the behaviour of other atmospheric constituents with horizontally-distributed near-surface sources.”

This means that the concentration of radon entirely depends on advection and mixing, over a flat ground the mixing depth will change at night with stability. The difference in the diurnal cycle between our sampling point (Lucas Heights) and Sydney is caused by drainage losses. Cool air that is high in radon will drain into the nearby valley. Meanwhile, the air that is “lost” through drainage, will be replaced from air higher up from the surface (which contains less radon). This is briefly explained in lines 272 to 277 as follows:

“Diurnal cycles of radon (see Figure 2) were typically characterised by a morning maximum (when atmospheric mixing was weakest) and a late afternoon minimum, when the atmospheric boundary layer (ABL) was deepest and most well-mixed. Peak nocturnal radon concentrations were a factor of 2-3 lower than reported elsewhere in the Sydney Basin [57], primarily attributable to this site’s location at the top of a ridge in complex topography (changes in elevation of ~180m within a 1 km radius of the site).”

Point 4: Generally, I would prefer that authors use IUPAP and IUPAC recommendations for writing units and quantities, for example: c(compound)/mol dm-3 or φ/ppb or t/°C etc.

Response 4: We agree with the reviewer that using standard recommendation from writing units make it easier to understand the analysed variables. We will apply this in further works However, given the short time provided to make the changes and that the units are not cause of confusion we decided to keep them as they are.

Reviewer 4 Report

The manuscript is devoted to the analysis of seasonal variation of biogenic and anthropogenic VOCs in a semi-urban area near Sydney. The study is relevant, the manuscript has a clear structure, and it is well illustrated. In general, I do not see any serious problems, except small inaccuracies that require correction.

In the title the comma is missing: “Sydney, Australia”.

Line 14: I suggest to clarify once in the abstract and once further in the main text of the manuscript that the studies were carried out from February (summer in the Southern Hemisphere) to June (winter in the Southern Hemisphere). This may be helpful for readers from the Northern Hemisphere.

Line 16: All studied biogenic and anthropogenic VOCs should be named.

It would be nice if the authors could shorten the abstract in order to keep only the most important results (maybe you can make the text shorter on lines 21-26 and on lines 27-34).

Lines 36-37: “The data is in submission process.” – this can be removed.

Lines 49-66: It has been repeated many times that VOCs are important precursors in the formation of ozone and fine particulate matter, that is, they play an essential role in atmospheric chemistry. It is necessary to describe these results in brief, indicating the most important findings.

Line 142: Please indicate that the survey was conducted from February to June 2019.

Line 151: The text has multiple reference source errors (“Error! Reference source not found. ”). You need to fix this issue throughout the entire manuscript.

Line 179: It is necessary to indicate which VOCs were studied.

Figure 1: there is no blue indicator in the figure (indicated in the caption below). What does the yellow indicator show?

Lines 182-265: Most of the text and tables from Methods section can be transferred to the Supplementary section. The methods should be presented very briefly in the main text.

Line 302: “window (1900h – 0500h) referenced to the 1900h value” => “window (19:00h – 05:00h) referenced to the 19:00h value”.

Table 4: symbols in the names of the columns are missing (instead of them there are “??” signs).

Discussion section is extremely long. It must be divided into subsections.

Line 379: High concentrations of AVOCs are found with northwest winds. In the manuscript, this is explained by the influence of Sydney, located 30 km from the sampling station. However, much closer (only 2 km) to the northwest of the sampling site is the municipal waste management facility (line 154). It is known that landfills can be active sources of VOCs (10.1016/j.envpol.2017.02.028; 10.1016/j.wasman.2018.05.013; 10.1016/j.envres.2020.110068 etc.). It is necessary to discuss this possibility. It is also not entirely clear why the BVOCs maxima are noted with northwest winds in summer (Supplementary Figure 4), if the main forest area is located south and southwest of the study point (as noted in Figure 1)?

Line 503: First word of the sentence is missing.

The manuscript requires minor revisions.

Author Response

We thank the referees for their time spent carefully reviewing the manuscript, and in their opinions regarding the science and presentation of the material, please see the attachment. In the attached file the referee’s comments are in black and the authors responses are in red.
